# Therapeutic targeting nudix hydrolase 1 creates a MYC-driven metabolic vulnerability

Minhui Ye[1,2,7], Yingzhe Fang[1,2,7], Lu Chen[2], Zemin Song[3], Qing Bao[2], Fei Wang[4], Hao Huang[2], Jin Xu[2], Ziwen Wang [5], Ruijing Xiao[3], Meng Han[6], Song Gao [5], Hudan Liu [2], Baishan Jiang [2] ✉ & Guoliang Qing [1,2,3] ✉

Tumor cells must rewire nucleotide synthesis to satisfy the demands of unbridled proliferation. Meanwhile, they exhibit augmented reactive oxygen species (ROS) production which paradoxically damages DNA and free deoxyribonucleoside triphosphates (dNTPs). How these metabolic processes are integrated to fuel tumorigenesis remains to be investigated. MYC family oncoproteins coordinate nucleotide synthesis and ROS generation to drive the development of numerous cancers. We herein perform a Clustered Regularly Interspaced Short Palindromic Repeats (CRISPR)-based functional screen targeting metabolic genes and identified nudix hydrolase 1 (NUDT1) as a MYC-driven dependency. Mechanistically, MYC orchestrates the balance of two metabolic pathways that act in parallel, the NADPH oxidase 4 (NOX4)-ROS pathway and the Polo like kinase 1 (PLK1)-NUDT1 nucleotide-sanitizing pathway. We describe LC-1-40 as a potent, on-target degrader that depletes NUDT1 in vivo. Administration of LC-1-40 elicits excessive nucleotide oxidation, cytotoxicity and therapeutic responses in patient-derived xenografts. Thus, pharmacological targeting of NUDT1 represents an actionable MYC-driven metabolic liability.

The hyperactive synthesis and use of nucleotide triphosphates (NTPs) and their deoxy counterparts (dNTPs) is a universal metabolic feature of tumor cells[1]. Both NTPs and dNTPs are rate-limiting for multiple essential biological processes during malignant transformation, including DNA replication and gene transcription[2]. Tumor cells frequently exhibit augmented ROS levels resulting from metabolic abnormalities and oncogenic activation. While moderate ROS introduces DNA mutations and facilitates tumor growth, when in excess, it paradoxically elicits cell death via damaging DNA and/or free dNTPs. In particular, the free dNTPs are 190–13,000 times more susceptible to oxidative damage in comparison to the genomic DNA[3]. Thus, an important question is how tumor cells couple these synthetic processes to sustain a delicate metabolic homeostasis vital for tumor cell survival.

The MYC family of oncoproteins, including MYC, MYCL, and MYCN, is a paradigmatic regulator of global metabolic reprogramming in a broad spectrum of human cancers[4,5]. MYC enforces hyperactive nucleotide synthesis by promoting the expression of de novo pathway enzymes and indirectly support the de novo pathway flux by enhancing cellular uptake of glucose and additional nutrients to provide the required ATP, ribose and one-carbon metabolites[5]. The presence of supraphysiological concentrations of nucleotide pools imposes an onerous cost on tumor cells to counteract their disastrous damages by intracellular ROS, whose levels are simultaneously augmented upon

[1]Department of Urology, Medical Research Institute, Zhongnan Hospital of Wuhan University, Wuhan University, Wuhan 430071, China. [2]Frontier Science Center for Immunology and Metabolism, Wuhan University, Wuhan 430071, China. [3]TaiKang Center for Life and Medical Sciences, School of Basic Medical Sciences, Wuhan University, Wuhan 430071, China. [4]School of Life Science and Technology, ShanghaiTech University, Shanghai 201210, China. [5]State Key Laboratory of Oncology in South China, Collaborative Innovation Center for Cancer Medicine, Sun Yat-sen University Cancer Center, Guangzhou 510060, China. [6]Protein Chemistry and Proteomics Facility, Tsinghua University Technology Center for Protein Research, Beijing 10084, China. [7]These authors contributed equally: Minhui Ye, Yingzhe Fang. ✉e-mail: baishan_jiang@whu.edu.cn; qingguoliang@whu.edu.cn

MYC hyperactivation. Therefore, an unresolved question is how key cellular processes underlying tumor cell growth, such as nucleic acid metabolism and ROS generation, become coordinated and are maintained, and whether this crosstalk reflects a unique vulnerability that could be targeted. This question is particularly critical, as the MYC family oncoproteins, at present, remain "undruggable"[6].

In this work, we discovered that MYC orchestrated the crosstalk of two parallel metabolic pathways, the NOX4-ROS pathway and the PLK1-NUDT1 nucleotide-sanitizing pathway, to promote tumor cell survival. These findings delineate a self-regulating circuitry through which tumor cells ensure balanced coordination of the production of ROS and the integrity of nucleotide pools. Importantly, by specifically inhibiting the expression of NUDT, we demonstrate synthetic lethality in MYC-overexpressing cells and dramatically decreased tumorigenic potential in vivo, illuminating an important vulnerability and therapeutic window for cancers driven by "undruggable" oncogenes, such as *MYC*.

## Results

### A CRISPR screen identifies NUDT1 as a MYC-driven metabolic dependency

As master regulators of metabolic reprogramming and redox homeostasis, the MYC family oncoproteins represent a paradigm to assess the functional relationship between metabolic processes in human cancer. As a start, we performed a CRISPR-based negative selection screen for metabolic genes whose loss potentiates cell death upon MYC hyperactivation. Such genes should reveal processes that help tumor cells adapt to and thrive on MYC-driven metabolic reprogramming. We stably expressed Cas9 in SHEP MYCN-ER cell line bearing a 4-hydroxytamoxifen (4-OHT)-activating *MYCN* transgene[7]. A functionally verified clonal cell line (Supplementary Fig. 1a–c), which we termed SHEP MYCN-ER-Cas9, was chosen for the screen.

We next constructed a focused sgRNA library comprising over 35,000 sgRNAs targeting 2745 metabolic enzymes and transporters as well as 500 control sgRNAs (Supplementary Fig. 1d). SHEP MYCN-ER-Cas9 cells were transduced with the pooled library and propagated for 14 population doublings in the presence or absence of 4-OHT (Fig.1a). Using massively parallel sequencing, we measured the abundances of all the sgRNAs in both vehicle and 4-OHT treated cells at the beginning and end of the culture period. A total of 189 genes were identified exhibiting >2-fold reduction ($p < 0.05$) in 4-OHT treated cells (Supplementary Data 1). The best scoring gene in the screen was *NUDT1* (Fig. 1b and Supplementary Fig. 1e). Indeed, the log-fold change score for each NUDT1 sgRNA showed a dramatic difference between 4-OHT and vehicle treatment (Fig.1c). A pathway analysis revealed that gene sets related to DNA protection and glutathione metabolic process were enriched in top depleted hits while those related to protein glycosylation and ceramide biosynthetic process were enriched in top accumulated ones (Supplementary Fig. 1f, g).

As a hydrolase, NUDT1 catalyzes the conversion of 8-oxo-dGTP to 8-oxo-dGMP that sanitizes damaged nucleotides and prevents the incorporation of oxidized dNTPs into DNA, which otherwise can result in disastrous DNA damage and cell death (Fig. 1d). We further verified NUDT1 as a unique metabolic dependency in MYC-overexpressing cells. shRNA knockdown of NUDT1 expression selectively induced SHEP MYCN-ER cell death upon 4-OHT induction of MYCN hyperactivation (Fig. 1e). NUDT1 depletion similarly elicited robust apoptosis in MYC-overexpressing Burkitt's lymphoma P493 cells, while administration of tetracycline (Tet) to repress MYC expression significantly inhibited cell death (Fig. 1f). We then extended our study to a panel of human cancer cell lines exhibiting differential MYC/MYCN levels (Fig. 1g). As expected, NUDT1 inhibition induced significant cell death in all the tumor cells associated with MYC/MYCN overexpression, whereas those with low MYC/

MYCN levels exhibited minimal cell death (Fig. 1h), underscoring the association between MYC overexpression and NUDT1 dependency. Enforced expression of wild-type (WT) NUDT1, but not the hydrolase-inactive mutant (E56A)[8], efficiently reversed the viability of *MYCN*-amplified Kelly cells (Fig. 1i), arguing that the catalytic activity of NUDT1 is required for tumor cells to adapt to MYC-driven metabolic reprogramming.

### NUDT1 counteracts the death-inducing nucleotide oxidation caused by MYC(N)-NOX4-ROS pathway activation

In addition to dramatically increasing the abundance of intracellular nucleotide pools (via activation of nucleotide synthesis, Supplementary Fig. 2a), a prominent function of MYC in metabolic reprogramming is augmenting ROS production[9,10] (Fig. 2a). Yet, the underlying mechanisms remains to be deciphered. A major endogenous source of superoxide and hydrogen peroxide is from the nicotinamide adenine dinucleotide phosphate (NADPH) oxidase (NOX) family of enzymes (NOX1–5, DUOX1,2)[11]. We postulate that oncogenic MYC might specifically activate the expression of these enzymes, enabling augmented ROS production. To discriminate which member(s) might be involved in, we analyzed their induction in SHEP MYCN-ER and P493 cells. Interestingly, 4-OHT induction of MYCN in SHEP cells selectively activated *NOX4* expression (Fig. 2b), with that of *NOX2–5* and *DUOX2* barely detected. Inactivation of MYC by Tet consistently diminished NOX4 induction in P493 cells (Supplementary Fig. 2b), confirming that MYC family oncoproteins specifically stimulate NOX4 expression in human tumor cells. Indeed, *MYCN* and *MYC* knockdown respectively inhibited NOX4 expression in Kelly and SF188 cells (Supplementary Fig. 2c, d). To confirm whether *NOX4* is a direct MYC target, we pretreated SHEP MYCN-ER cells with cycloheximide (CHX) to shut down global protein translation before a time course induction of MYCN by 4-OHT. Chromatin immunoprecipitation (ChIP) assays revealed that MYCN was selectively recruited to the E box region of the *NOX4* gene upon 4-OHT induction in a time-dependent manner, proportionally leading to increased *NOX4* mRNA levels (Fig. 2c and Supplementary Fig. 2e, f), supporting that MYCN directly activated *NOX4* transcription to elevate ROS production. We then analyzed NOX4 protein levels in multiple tumor cells and primary tumor samples, and found NOX4 expression was markedly elevated in MYC(N)-high tumor cells and tumor samples (Supplementary Fig. 2g). To evaluate whether this observation is representative of what occurs in human tumors, we analyzed neuroblastomas with and without *MYCN* amplification. *NOX4* mRNAs were significantly elevated in the *MYCN*-amplified tumors when compared with nonamplified ones (Supplementary Fig. 2h). Moreover, expression between *MYCN* and *NOX4* is significantly correlated in *MYCN*-amplified neuroblastoma samples (Supplementary Fig. 2i). All these data support that MYC(N)-induced NOX4 upregulation occurs in human tumors. As a functional support, depletion of NOX4 expression significantly inhibited MYCN induction of ROS accumulation but barely affected the basal ROS levels (Fig. 2d), while overexpression of NOX4 further increased ROS production, supporting that NOX4 plays a critical role in MYCN augmentation of ROS generation (Supplementary Fig. 2j).

The implication of NUDT1 in sustaining viability of MYC(N)-overexpressing cells prompted us to hypothesize that they may rely on NUDT1 to counteract the overwhelming accumulation of lethally oxidized nucleotides resulting from NOX4-ROS pathway activation. Indeed, *NUDT1* depletion caused a significant accumulation of 8-oxo-dG, a paradigm of oxidized nucleotide, upon MYCN hyperactivation in SHEP MYCN-ER cells, as evidenced by avidin immunofluorescence (Fig. 2e, f). Overexpression of 8-oxoguanine DNA glycosylase OGG1, which removes 8-oxo-dG in DNA[12], was used as a control for avidin readout. Notably, N-Acetyl-L-cysteine (NAC), a widely used ROS scavenger, substantially counteracted the toxic nucleotide oxidation caused by NUDT1 depletion (Fig. 2e, f), and effectively rescued cell

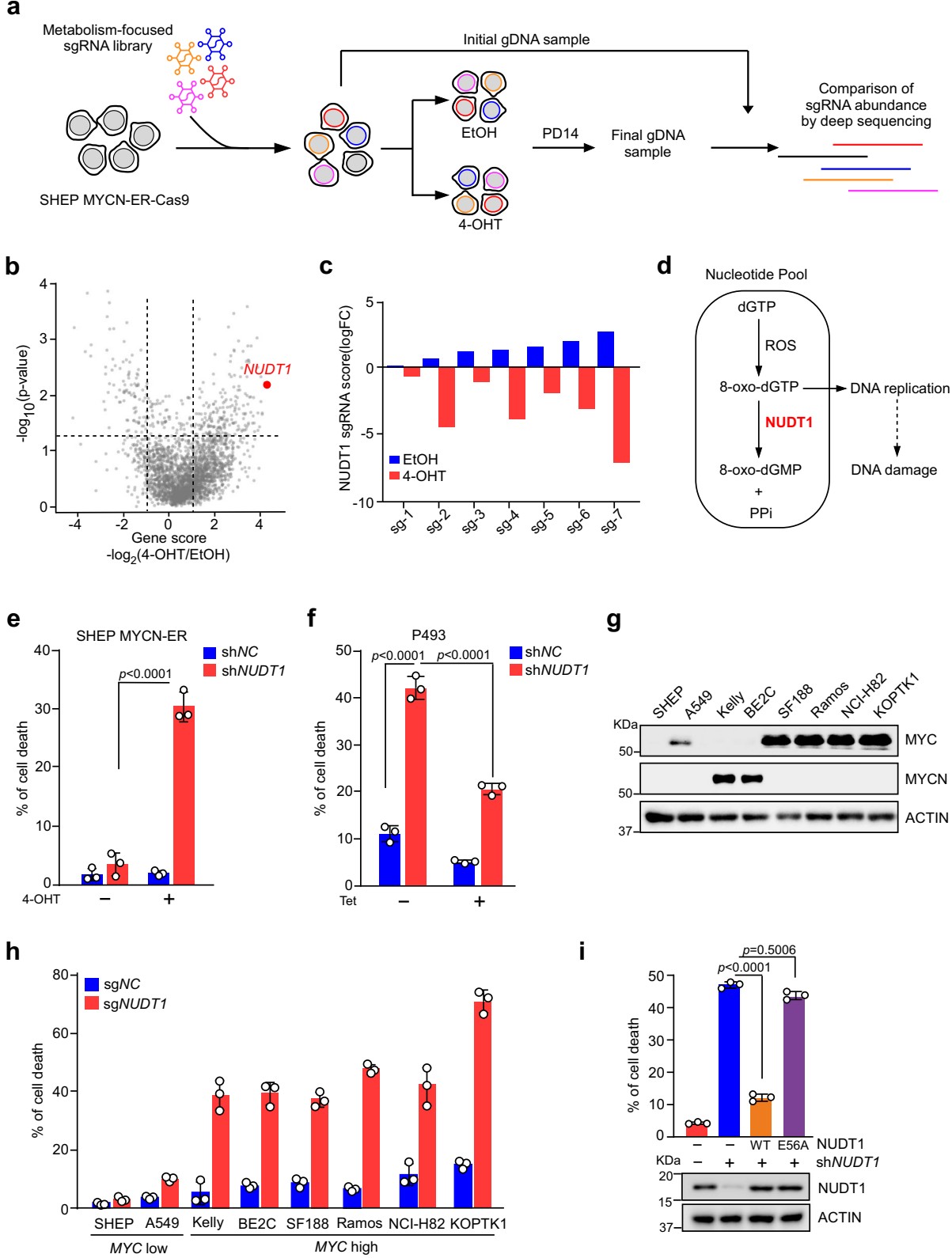

death (Fig. 2g). Moreover, inhibition of *NOX4* in *MYCN*-amplified Kelly cells or *MYC*-amplified SF188 cells similarly rescued NUDT1-depeltion induced cell death (Fig. 2h, i and Supplementary Fig. 2k, l), while ectopic expression of NOX4 increased the cell death further (Supplementary Fig. 2m). All these findings argue that NUDT1 is essential for detoxifying the lethally damaged nucleotides due to MYC hyper-activation of the NOX4-ROS axis.

## MYC(N)-induced PLK1 directly phosphorylates NUDT1 and enhances its enzymatic activity

Given MYC(N)-overexpressing tumor cells strictly rely on NUDT1 to sanitize oxidized nucleotides for viability, we postulated that NUDT1 expression and/or catalytic activity should be enforced to adapt to MYC(N)-driven metabolic reprogramming. Interestingly, when normalized to the same amount of whole-cell lysates, the

**Fig. 1 | CRISPR-Cas9 functional genomics screen identifies NUDT1 as a selective MYC-driven metabolic dependency. a** Schematic depicting the pooled CRISPR-based screen. SHEP MYCN-ER-Cas9 cells were transduced with a metabolism-focused library of lentiviral sgRNAs targeting 2745 metabolic genes. At population doubling 0 (PD 0), cells were cultured with or without 4-OHT treatment (50 nM) for 14 population doublings (PD 14). Relative abundance of sgRNAs in PD 14 was compared to those in PD 0. **b** Gene scores in untreated versus 4-OHT treated SHEP MYCN-ER cells. The gene score means the median $log_2$ fold change of all sgRNAs abundance targeting each metabolic gene during the culture period. The analysis was performed utilizing MAGeCK (version 0.5.9.5), which involved the calculation of $p$-values. **c** Abundance changes in the primary pooled screen of the individual *NUDT1* sgRNAs with or without 4-OHT treatment. **d** The biochemical reaction catalyzed by NUDT1. NUDT1 hydrolyzes the representative oxidized triphosphates 8-oxo-dGTP into 8-oxo-dGMP, which is unable to incorporate into DNA. **e** Cell death analysis of SHEP MYCN-ER cells treated with 4-OHT (200 nM) for 24 h and infected with the *NUDT1* shRNA or control (sh*NC*) lentivirus particles. **f** Cell death analysis of P493 cells treated with Tetracycline (Tet, 100 ng/ml) for 24 h and infected with the *NUDT1* shRNA or control (sh*NC*) lentivirus particles. **g** Immunoblots of MYCN and MYC in various tumor cells, with β-Actin as a loading control. **h** Cell death analysis of various tumor cells infected with the *NUDT1* sgRNA or control (sg*NC*) lentivirus particles. **i** Cell death analysis of *NUDT1*-depleted Kelly cells expressing NUDT1 wild-type (WT) or the catalytically dead mutant (E56A). **e**, **f**, **h**, **i** data are shown as averages of technical triplicates; **e–i** these experiments were independently repeated three times with similar results. Statistical significance was determined by two-way ANOVA (**e**, **f**, **i**). Source data are provided as a Source Data file.

NUDT1 catalytic activity significantly increased in SHEP MYCN-ER cells upon 4-OHT treatment over time even though its protein abundance remained largely unchanged (Fig. 3a, b), arguing that the markedly increased enzyme activity is not due to NUDT1 mass accumulation.

We surmised that the expression of particular proteins, which bind NUDT1 and modulate its activity via protein modifications, was specifically induced by MYC family oncoproteins. To probe this possibility, we respectively performed immunoprecipitation assays using the SHEP cellular extracts with or without 4-OHT treatment. LC–MS (liquid chromatography tandem mass spectrometry) analysis of NUDT1-bound immunoprecipitates revealed that PLK1 emerged as the top, high-confidence hit in 4-OHT treated cells (Fig. 3c). Of note, we have previously shown that an important role of MYC family oncoproteins is to function as potent activators of PLK1 expression in human cancers[13]. Indeed, both the total and kinase active PLK1 (revealed by PLK1 T210 phosphorylation) levels were significantly elevated in SHEP MYCN-ER cells upon 4-OHT administration (Supplementary Fig. 3a), raising an interesting possibility that PLK1 functions as an ideal nexus between MYC(N) and NUDT1. In support of this notion, interaction between endogenous PLK1 and NUDT1 was markedly increased in 4-OHT treated SHEP cells (Fig. 3d). Moreover, a much stronger PLK1-NUDT1 interaction was observed in Kelly (*MYCN*-amplified) and SF188 (*MYC*-amplified) cells as compared to that in SHEP cells (Supplementary Fig. 3b). GST pull-down with recombinant PLK1 and NUDT1 validated a direct interaction between NUDT1 and PLK1 (Fig. 3e).

These findings prompted us to surmise that PLK1 directly phosphorylates NUDT1 and enhances its catalytic activity. Of note, the constitutively active PLK1 (PLK1-T210D), but not its kinase inactive form (PLK1-K82R), gave rise to a specific phosphorylation signal in recombinant NUDT1 protein (Fig. 3f, compare lanes 5−6 vs 2−3). More importantly, co-transfection of NUDT1 with PLK1-T210D (kinase active) significantly increased its hydrolase activity while co-transfection with PLK1-K82R (kinase inactive) did not (Fig. 3g), arguing that PLK1-directed phosphorylation is crucial for NUDT1 catalytic activity. In vitro, kinase assays coupled with LC−MS analysis revealed three potential PLK1 phosphorylation sites (NUDT1 S121, T144, and T155) (Supplementary Fig. 3c–e). We constructed the corresponding non-phosphorylatable mutants (S121A, T144A, and T155A) and repeated the in vitro kinase assay with $\gamma$-$^{32}$P-ATP. Interestingly, substitution of serine 121 to alanine, but not the other two, in NUDT1 abolished the phosphorylation signal by PLK1 (lane 6), demonstrating that this residue is the major PLK1 phosphorylation site (Fig. 3h). We then generated a rabbit polyclonal antibody against NUDT1 S121 phosphorylation (Supplementary Fig. 3f). As expected, MYCN activation in SHEP MYCN-ER cells gave rise to a potent phosph-S121 signal, which was abrogated upon incubation of the cell lysates with calf-intestinal alkaline phosphatase (Fig. 3i). Consistently, we confirmed that the NUDT1 S121 phosphorylation was strictly dependent on PLK1 kinase activity in both Kelly and SF188 cells (Fig. 3j and Supplementary Fig. 3g). Moreover, we

observed elevated p-S121 NUDT1 levels in MYC(N) high tumor cells and tumor samples in comparison to those with low MYC(N) expression (Supplementary Fig. 3h).

We next reconstituted in vitro assays to directly monitor the enzymatic activities of wild-type NUDT1 and corresponding mutant S121D. Compared to WT NUDT1, the S121D mutant exhibited an increased catalytic activity as manifested by kinetic parameters of Vmax, Km, and Kcat/Km (Fig. 3k, l). Of note, compared with WT NUDT1, ectopic expression of the S121A mutant in SHEP MYCN-ER cells can only partially reduced the 8-oxo-dG levels upon depletion of endogenous NUDT1 (Supplementary Fig. 3i, j), supporting the contribution of enhancing NUDT1 catalytic function by PLK1-mediated phosphorylation. Based on the ternary structure of NUDT1 and 8-oxo-dGTP complex available (Protein Data Bank code 5FSI), S121 is a surface-exposed residue located to one of the substrate binding loops (Supplementary Fig. 3k). In the same loop, the invariant W117, D119 and D120 are responsible for coordinating the incoming nucleotide via the interaction with its guanine base. Phosphorylation of S121 is likely to affect the configuration and flexibility of the substrate binding loop, which may accelerate the product-substrate exchange, and thereby enhancing the turnover of 8-oxo-GTP. Altogether, these findings support that MYC(N) promotes NUDT1 catalytic activity via enforcing PLK1-directed NUDT1 phosphorylation.

## *Nudt1* knockout impedes MYC-driven tumor growth
We constructed *Nudt1*$^{null}$ mice (Fig. 4a–c) and employed them to genetically address the requirement of NUDT1 activity in MYC(N)-driven tumorigenesis. In agreement with a previous report[14], mice homozygous null for the *Nudt1* gene are viable, fertile, and display no gross phenotypic abnormalities (Supplementary Fig. 4a–k). We then crossed *Nudt1*$^{null}$ mice with TH-*MYCN* transgenic mice[15] (Fig. 4d), which faithfully recapitulate the pathogenesis of *MYCN*-amplified neuroblastomas in human. Primary sympathetic ganglia from TH-*MYCN*$^{+/+}$*Nudt1*$^{null}$ mice were isolated along with age-matched controls. As expected, *Nudt1* knockout significantly increased the expression of phosphorylated γH2AX in sympathetic ganglia associated with elevation of cleaved-Caspase 3 (Fig. 4e), corroborating that *NUDT1* deletion caused overwhelming DNA damages and cell death in MYCN-driven tumor contexts in vivo. Indeed, TH-*MYCN*$^{+/+}$*Nudt1*$^{null}$ mice displayed a remarkable delay in tumor development compared to their TH-*MYCN*$^{+/+}$ littermates, with approximately 60% of TH-*MYCN*$^{+/+}$*Nudt1*$^{null}$ mice that lived beyond 100 days (and 5 mice, having been tumor-free for >850 days, are still alive), a time point at which all TH-*MYCN*$^{+/+}$ mice had died (Fig. 4f). Consistently, immunohistochemistry (IHC) analysis showed much more 8-oxo-dG, γH2AX and cleaved-Caspase 3 staining in tumor sections from TH-*MYCN*$^{+/+}$*Nudt1*$^{null}$ mice (Fig. 4g, h).

To extend our observations in TH-*MYCN*$^{+/+}$ mice, we next assessed NUDT1 in a T-cell acute lymphoblastic leukemia (T-ALL) model, where MYC is found overexpressed[16]. Lineage-negative (Lin$^-$) bone marrow cells from either *Nudt1* WT or null mice were

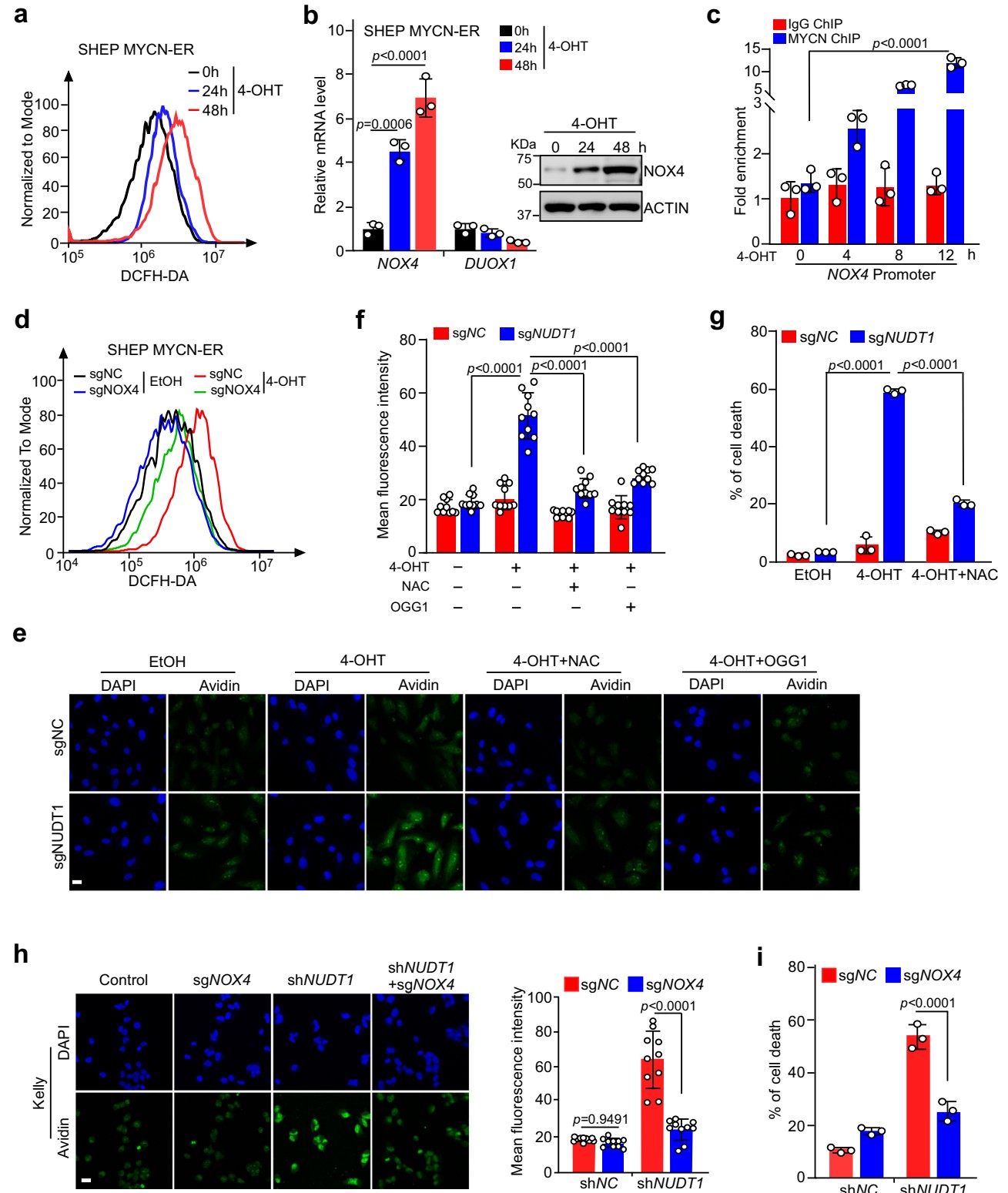

transduced with the MSCV-IRES-GFP retroviral vector expressing intracellular NOTCH1 (ICN1), and then transplanted into irradiated recipient mice (Supplementary Fig. 5a). Mice receiving ICN1-expressing *Nudt1* WT cells exhibited a rapid increase in the proportion of GFP+ leukemia cells in their peripheral blood and developed overt CD4+CD8+ T-ALL within six weeks (Supplementary Fig. 5b–f). In sharp contrast, *Nudt1* deletion significantly delayed leukemia onset and extended the survival of transplanted mice (Supplementary Fig. 5g). All these data support that NUDT1

expression is required to sustain tumor cell survival in both mouse and human cancers that rely on MYC(N) hyperactivation.

## Development of selective NUDT1 degraders

To investigate the therapeutic benefit whether loss of function of NUDT1 creates a MYC(N)-driven vulnerability, we employed proteolysis targeting chimera (PROTAC)-based technology to pharmacologically degrade NUDT1 protein, which in principle phenocopies the effects of gene knockout[17]. We noticed that C4 Therapeutics disclosed

**Fig. 2 | The MYC-NOX4 axis promotes ROS generation and nucleotide oxidation. a** Time course analysis of ROS levels in 4-OHT (200 nM) treated SHEP MYCN-ER cells. One million cells were stained with DCFH-DA (5 μM) for 20 min at 37 °C, followed by detection with flow cytometry. **b** Real-time qPCR analysis and immunoblot of NOX4 in SHEP MYCN-ER cells upon 4-OHT treatment (200 nM). **c** ChIP-qPCR analysis of MYCN binding to the *NOX4* promoter in SHEP MYCN-ER cells. **d** Detection of ROS levels in SHEP MYCN-ER cells transduced with the *NOX4* sgRNA and treated with 4-OHT (200 nM) for 48 h. **e** Representative avidin immuno-fluorescence images in SHEP MYCN-ER cells. Cells were transduced with the *NOX4* sgRNA and treated with 4-OHT (200 nM) in the presence or absence of NAC (100 μM), or OGG1 overexpression. Scale bar, 20 μm. **f** Quantification of 8-oxo- dGTP incorporation in SHEP MYCN-ER cells (*n* = 10 images) as shown in (**e**). Data are means ± SD. **g** Cell death analysis of SHEP MYCN-ER cells treated as shown in (**e**). **h** Representative avidin immunofluorescence images in Kelly cells transduced with the *NUDT1* shRNA and *NOX4* sgRNA lentivirus particles. Scale bar, 20 μm. Quanti-fication of immunofluorescence signals are shown on the right (*n* = 10 images). Data are means ± SD. **i** Death analysis of Kelly cells transduced with the *NUDT1* shRNA and *NOX4* sgRNA lentivirus particles. **b**, **c**, **g**, **i** data are shown as averages of tech-nical triplicates, and these experiments were independently repeated three times with similar results; **e**, **h** these experiments were independently repeated twice with similar results. Statistical significance was determined by one-way ANOVA (**b**) or two-way ANOVA (**c**, **f**–**i**). Source data are provided as a Source Data file.

three selective, potent NUDT1 PROTAC degraders during the course of our study[18]. However, these degraders were rapidly metabolized in vivo, with half-life values < 0.8 h in pharmacokinetic (PK) studies. To develop potent NUDT1 degraders with better PK properties, we rationally designed and synthesized a series of bifunctional degraders by conjugating the NUDT1 binder derived from compound 32[18] to a racemic cereblon (CRBN) ligand Thalidomide[19] with rigidified linkers (Fig. 5a, b). Among these degraders, LC-1-40 was identified as one of the best candidates due to its high efficacy in degrading NUDT1 in SHEP MYCN-ER model system with a half-maximal degradation con-centration (DC$_{50}$) value of 0.97 nM, and a maximum degradation (Dmax) value of 96% at a drug dose of 50 nM (Fig. 5c, d). Moreover, administration of LC-1-40 induced a dose- and time-dependent NUDT1 degradation (Fig. 5c, d).

Treatment of Kelly and SF188 cells with LC-1-40 caused minimal changes in *NUDT1* mRNA levels (Supplementary Fig. 6a). Of note, administration of MG132 (a specific 26 S proteasome inhibitor), TH287 (a previously identified NUDT1 inhibitor[20]) or Pomalidomide (a CRBN ligand) efficiently rescued LC-1-40 induced NUDT1 loss (Fig. 5e, f and Supplementary Fig. 6b), arguing that LC-1-40 is a *bone fide* CRBN- and proteasome-dependent, NUDT1-targeting PROTAC. To further evaluate LC-1-40 selectivity, we performed a quantitative proteomic analysis. Of particular note, the results showed that NUDT1 stood out as the most downregulated protein in SF188 cells treated with LC-1-40 for 6 h (Fig. 5g). To further validate the LC-1-40 specificity, we generated an E77K dormant mutant, which retains the NUDT1 enzymatic activity but is resistant to LC-1-40 degradation (incapable of NUDT1 binding) (Fig. 5h, i). As expected, the E77K mutant rescued LC-1-40 induced cell death as effectively as the WT NUDT1 (Fig. 5j). All these data demonstrated that LC-1-40 is a highly potent, on-target NUDT1 degrader.

To assess the therapeutic potential of LC-1-40 in vivo, we eval-uated its metabolic stability in male C57BL/6 mice (Fig. 5k). A PK study was conducted following a single dose administration in mice intra-venously (IV, 5 mg/kg), intraperitoneally (IP, 30 mg/kg), and orally (PO, 30 mg/kg). LC-1-40 displayed high plasma concentrations, achieving a Cmax of 4373 ng /mL, 3080 ng/mL, and 1923 ng/mL, and exhibited great metabolic stability in plasma with a half-life value of 6.34 h, 42.7 h, and 7.13 h, respectively. More importantly, the plasma con-centrations of LC-1-40 remained over 200 nM even at 24 h post-dosing, demonstrating its considerable in vivo stability. Of Note, LC-1-40 exhibited an oral bioavailability of 14.2%, nominating it as a promising lead for further optimization. All these data support that LC-1-40 is a selective NUDT1 degrader suitable for in vivo study.

### LC-1-40 elicits MYC(N)-driven synthetic lethality
To evaluate whether LC-1-40 can elicit MYC(N)-dependent cell death, we applied SHEP MYCN-ER model system. As expected, administration of LC-1-40 markedly increased 8-oxo-dG levels and selectively induced SHEP cell death upon 4-OHT induction of MYCN hyperactivation, which was efficiently reversed by NAC (Fig. 6a and Supplementary Fig. 7a, b). Given the predominant role NUDT1 plays in MYC(N)-induced cell death, we speculate that the overwhelming nucleotide

and DNA damages induced by NUDT1 depletion upon MYC(N) acti-vation overrides the cell's capacity to repair. To examine this, we performed a time course experiments in SHEP MYCN-ER cells by administration of the NUDT1 degrader LC-1-40. As expected, when MYCN is inactive (4-OHT absent), administration LC-1-40 caused an initial increase in avidin and γH2AX staining in SHEP cells, but these signals gradually decreased as the treatment continued to 24 and 36 h. In sharp contrast, avidin and γH2AX staining continued to increase upon 4-OHT induction of MYCN hyperactivation (Supplementary Fig. 8a, b), leading to accumulation of lethal damages and ultimate cell death (Supplementary Fig. 8c). We further confirmed the MYC(N) dependency in a panel of tumor cell lines and primary tumor cells manifesting differential MYC(N) levels (Fig. 6b, c). These in vitro results suggested a functional interplay among NUDT1, MYC(N), and the MYC(N)-enforced oncogenic phenotypes that can be pharmacologi-cally targeted by LC-1-40. To test this, we established patient-derived xenografts using *MYCN*-amplified T409 and *MYCN*-nonamplified T423 primary neuroblastoma tumors (Fig. 6d). A single dose of LC-1-40 (IP, 30 mg/kg) caused a remarkable intratumoral NUDT1 depletion within 1 h (Fig. 6e), showing an excellent efficacy in vivo. Consistent with in vitro observations, LC-1-40 selectively impeded the growth of *MYCN*-amplified T409 tumors whereas exhibited minimal effect on that of *MYCN*-nonamplified T423 ones even though similar NUDT1 depletion was achieved (Fig. 6f–i and Supplementary Fig. 9a, b). Immunohisto-logical analysis showed LC-1-40 led to much more intratumoral nucleotide damage and apoptosis in *MYCN*-amplified T409, as quan-tified by 8-oxo-dG and cleaved-Caspase 3 staining (Supplementary Fig. 9c, d). LC-1-40 therapy was well tolerated by the evidence of body weight sustainability (Supplementary Fig. 9e, f). All of these results support the clinical potential of LC-1-40 as a lead for cancer ther-apeutics against MYC(N)-driven tumors.

## Discussion
The rewiring of cellular metabolism represents a major determinant of tumor progression and is considered as a hallmark of cancer. When duplicating mass, tumor cells must finely balance the divergent metabolic requirements to sustain cell function and survival[21–24]. How divergent metabolic processes are integrated to fuel tumor growth remains to be investigated.

MYC family of oncoproteins represents a paradigm to interrogate this important question, as these oncogenic transcription factors are deregulated in >50% of human cancers and reprogram many aspects of cell metabolism[5]. The results we present herein advance our under-standing of molecular mechanisms underlying metabolic crosstalk during malignant transformation. The key concept emerging from our data is that MYC finely coordinated two metabolic processes that act in parallel, the NOX4-ROS pathway and the PLK1-NUDT1 nucleotide-sanitizing pathway, to maintain tumor cell survival (Fig. 7). In aggre-gate, these findings unravel a MYC-directed mechanism whereby tumor cells ensure balanced coordination of the production of ROS and the sanitation of nucleotide pools.

The significant progress made toward understanding how parti-cular nutrients fuel tumor cell metabolism has rekindled immense

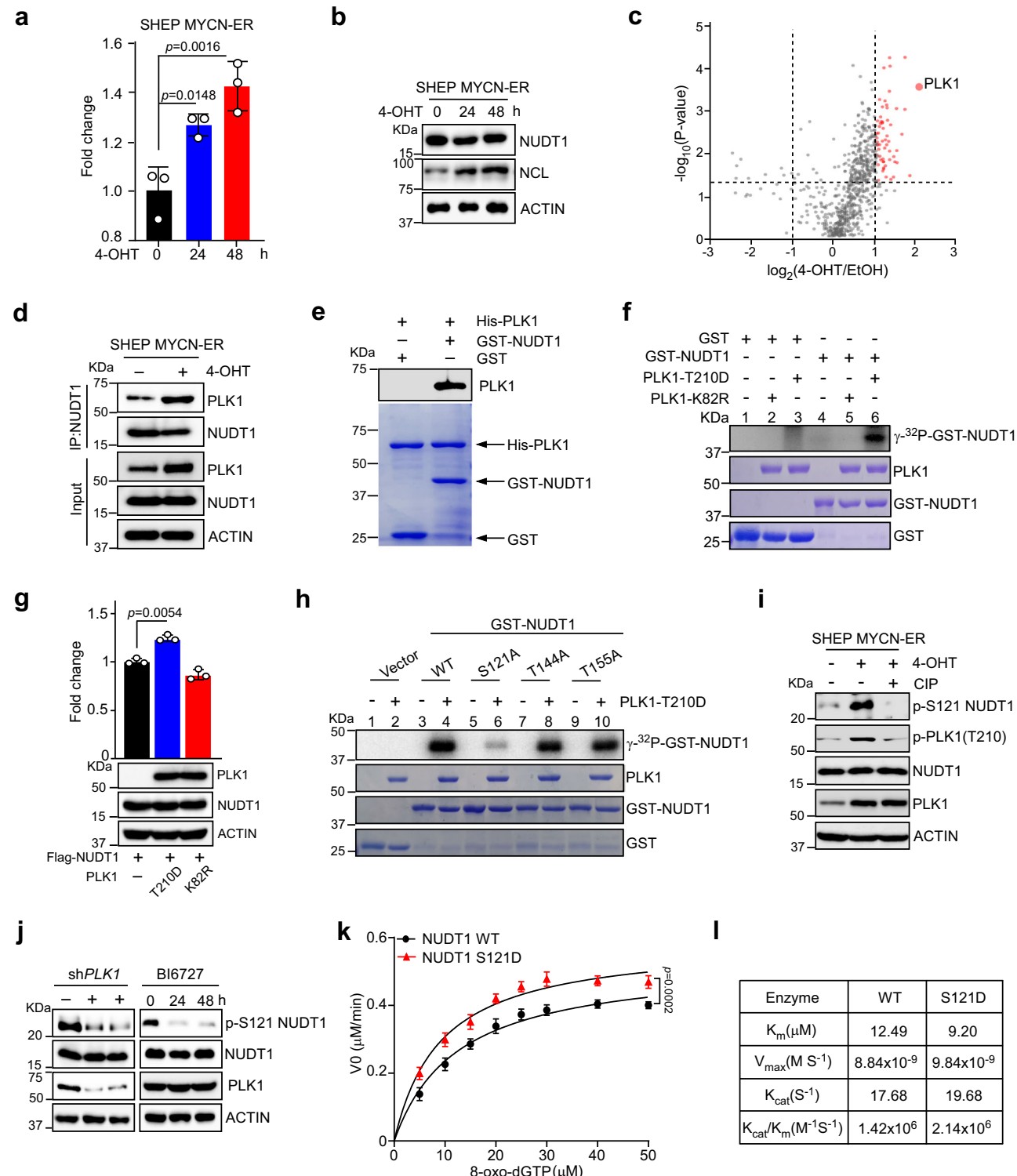

enthusiasm in development of inhibitors targeting metabolic enzymes as cancer therapeutics[25]. Yet, metabolism is often viewed as a housekeeping function for cells, and normal proliferating cells usually have similar metabolic requirements as cancer cells[23]. As such, in addition to the antimetabolite chemotherapies (non-tumor selective inhibitors of nucleotide metabolism), very few molecules that target central metabolism can enter clinical trials to achieve desirable efficacy[26]. Thus, a better understanding of the underlying tumor-centric metabolic alterations would enable the development and optimization of more effective strategies that therapeutically target cancer cell metabolism.

Deregulated redox homeostasis is a general phenotype of many cancers[27]. Augmented ROS levels in proliferating tumor cells can cause damage to DNA and free dNTPs, and trigger tumor cell death[28]. Removal of oxidized nucleotides by NUDT1 (also known as MTH1) might relieve tumor cells from replicative stress and thereby represent a vulnerability and an ideal target for anticancer compounds. Indeed, two previous investigations have posited NUDT1 as a universal target for cancer therapies based on the observations that knockdown of NUDT1 expression or small chemical inhibition of its activity selectively induced tumor cell death without affecting

**Fig. 3 | MYC-induced PLK1 phosphorylates NUDT1 and enhances its enzymatic activity. a, b** Assessments of NUDT1 enzymatic activity (**a**) and protein expression (**b**) in SHEP MYCN-ER cells exposed to 4-OHT (200 nM) for indicated time points. NCL (nucleolin) was used as a positive control to monitor MYCN activation. **c** Volcano plots showing NUDT1-interacting proteins in SHEP MYCN-ER cells with or without 4-OHT treatment (200 nM) for 24 h, analyzed using DEP (version 1.6.1). Proteins significantly enriched from three independent experiments with a fold change >2 and $p < 0.05$ compared to the control group are displayed as red dots. **d** Co-immunoprecipitation (Co-IP) to detect interaction between endogenous NUDT1 and PLK1 in SHEP MYCN-ER cells with or without 4-OHT treatment (200 nM) for 24 h. **e** GST pull-down to identify the direct interaction between NUDT1 and PLK1. Anti-His antibody was used to detect PLK1. **f** In vitro kinase assay of GST-NUDT1. Human PLK1-T210D or K82R purified from 293 T cells were used for the kinase assays and NUDT1 phosphorylation, with GST as a negative control, was detected by autoradiography (top panel). Loading controls were visualized by Coomassie blue staining (bottom panels). **g** Analysis of Flag-NUDT1 enzymatic

activity co-expressed with exogenous PLK1-T210D or K82R in 293 T cells. **h** In vitro kinase assay of GST-NUDT1 or the indicated mutants as shown in (**f**). **i** Immunoblots of NUDT1 p-S121 in SHEP MYCN-ER cells with or without 4-OHT treatment (200 nM). A portion of 4-OHT treated cell lysates were treated with calf-intestinal alkaline phosphatase (CIP) for 30 min at 37 °C before immunoblot. **j** Immunoblot of NUDT1 p-S121 in Kelly cells treated with PLK1 shRNA or BI6727 (20 nM). **k** Enzymatic activity assay of recombinant GST-NUDT1 or indicated mutants over a range of 8-oxo-dGTP concentrations. V0, reaction velocity. **l** Enzymatic characterization of GST-NUDT1 or its mutants toward 8-oxo-dGTP. The Michaelis–Menten equation was applied to saturation curves using the GraphPad Prism software and kinetic parameters were calculated. **a, g** data are shown as averages of technical triplicates; **k** data are shown as averages of technical quadruplicates; **a, b, d–k** these experiments were independently repeated three times with similar results. Statistical significance was determined by one-way ANOVA (**a**), two-way ANOVA (**g, k**) or two-tailed Student's *t*-test (**c**). Source data are provided as a Source Data file.

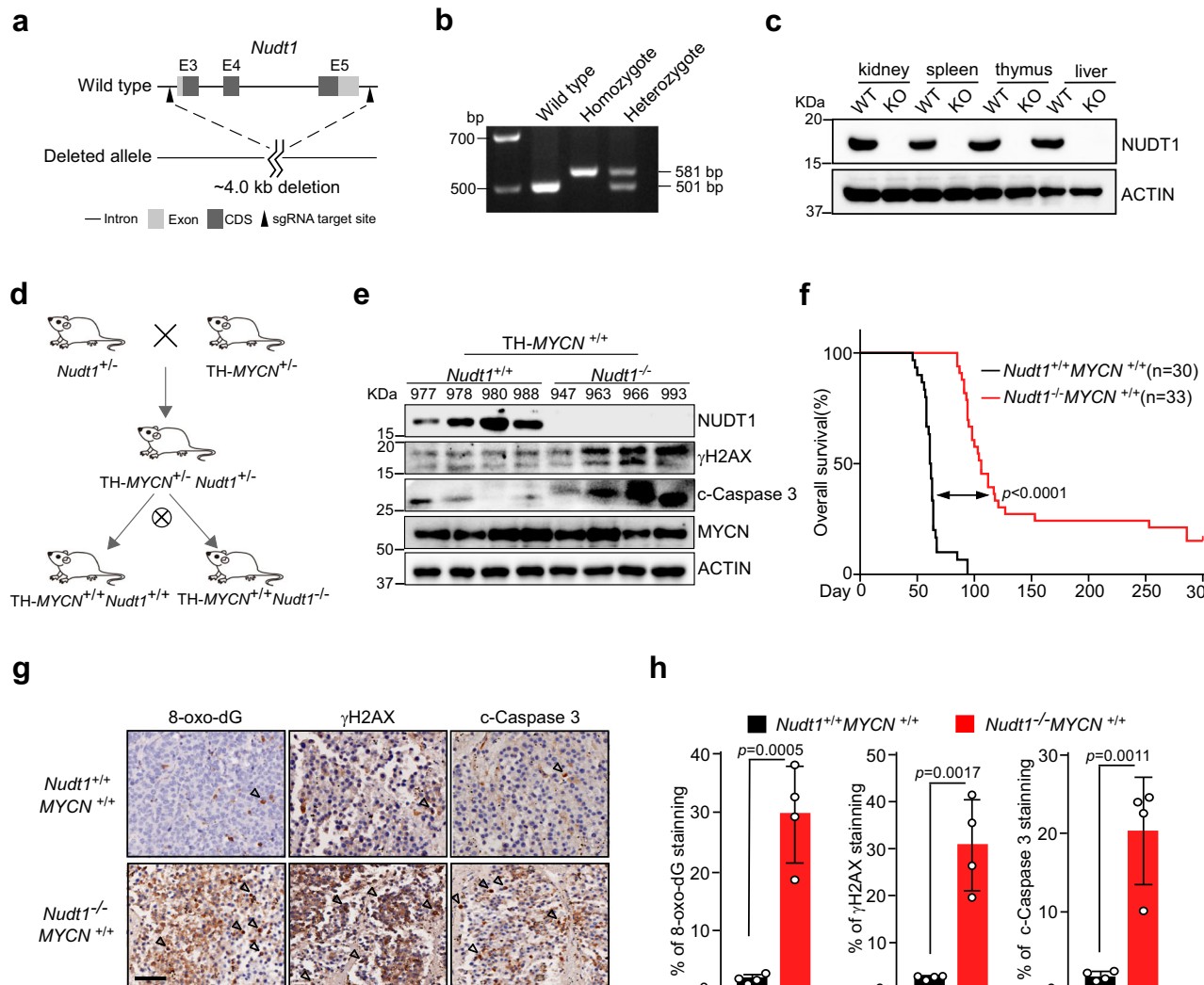

**Fig. 4 | *NUDT1* knockout suppresses *MYCN*-driven neuroblastoma. a** Targeting strategy for *Nudt1* allele knockout in mice by CRISPR-Cas9. **b** Genotype analysis of *Nudt1* knockout mice using PCR. PCR products clearly distinguish wild-type (single smaller band), homozygous (single larger band), and hemizygous (both bands) DNA fragments. **c** Immunoblot of NUDT1 in various organ lysates from *Nudt1⁻/⁻* mice and age-matched controls, with β-Actin as a loading control. **d** Breeding scheme for knockout of *Nudt1* alleles in TH-*MYCN* transgenic mice. **e** Immunoblots of indicated proteins in primary sympathetic ganglia lysates from *Nudt1⁻/⁻ MYCN⁺/⁺* and age-matched *Nudt1⁺/⁺ MYCN⁺/⁺* mice. Four mice were analyzed per group.

**f** Kaplan–Meier survival curve of TH-*MYCN⁺/⁺* mice with or without *Nudt1* knockout. Significance was determined by log-rank test. **g, h** Representative histological images of 8-oxo-dG, γH2AX and c-Caspase 3 staining in paraffin-embedded tumor sections from TH-*MYCN⁺/⁺* mice with or without *Nudt1* knockout (**g**). Scale bar, 50 µm. Triangles denote positive-stained cells. Histological stain was quantified using ImageJ and plotted in (**h**). Data shown represent the means of four tumor samples (±SD). **b, c** these experiments were independently repeated three times with similar results. Statistical significance was determined by unpaired two-tailed Student's *t*-test (**h**). Source data are provided as a Source Data file.

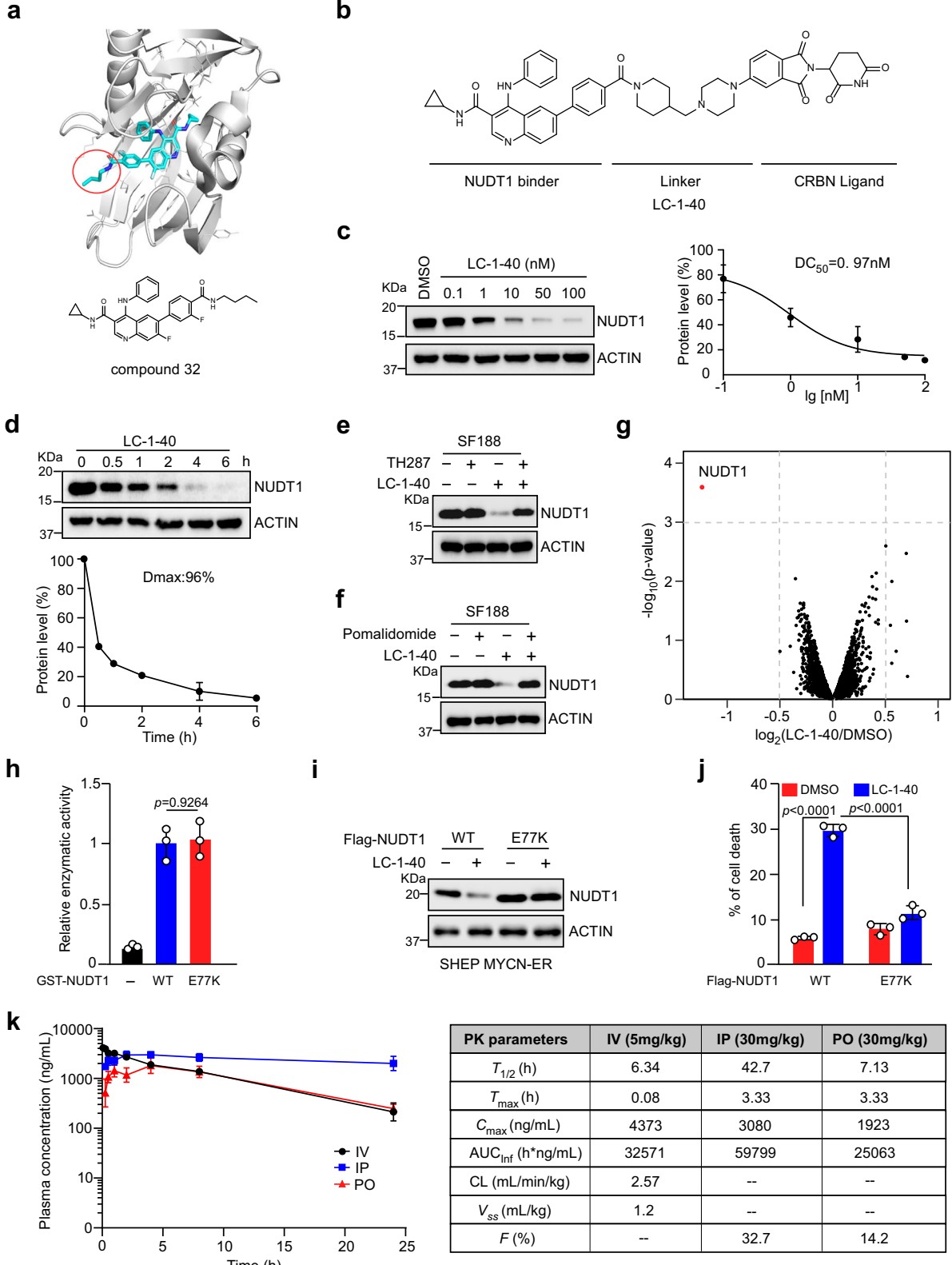

normal cells[20,29]. However, later independent studies failed to reproduce these results (arguing that the prior observations could be due to the off-target effects of NUDT1 siRNAs and/or small chemical inhibitors), challenging the notion of NUDT1 as a feasible, universal target for cancer therapy[30–32]. Therefore, whether NUDT1 can be an actionable therapeutic target remains controversial.

Here we performed a CRISPR-based functional screen targeting metabolic genes and, interestingly, identified NUDT1 as a MYC(N)-driven metabolic dependency. Mechanistic studies using a combination of genetic, pharmacologic, and biochemical approaches confirmed the underlying mechanism for this dependence involving MYC(N)-induced PLK1 to phosphorylate NUDT1 which, in turn, enhances NUDT1 hydrolase activity needed for sanitation of excessive

**Fig. 5 | LC-1-40 is a specific and potent NUDT1 degrader. a** Co-crystal structure of Compound 32 (cyan) bound to NUDT1 (gray, PDB:7N13) revealing solvent-exposed amide (circled) as an exit vector for linker attachment. **b** Structure of LC-1-40. **c** Representative immunoblot of NUDT1 in SHEP MYCN-ER cells treated with the indicated concentration of LC-1-40 for 1 h. Relative NUDT1 levels were quantified and plotted based on averages of three independent experiments. **d** Time-dependent NUDT1 depletion by LC-1-40 (50 nM) in SHEP MYCN-ER cells. Relative NUDT1 levels were quantified and plotted based on averages of three independent experiments. **e**, **f** NUDT1 immunoblots using SF188 cells pretreated with TH287 (2 μM) (**e**) or pomalidomide (10 μM) (**f**) for 2 h, and then subjected to an additional 6 h treatment with DMSO or LC-1-40 (50 nM). **g** Quantitative proteomics showing relative abundance of proteins in SF188 cells treated with 100 nM LC-1-40 for 6 h. The analysis was performed by the DEP software (version 1.6.1), which was also used to calculate the *p*-values, based on SF188 samples from three independent experiments. **h** Assessment of NUDT1 (WT and the E77K mutant) enzymatic activity. Data shown as averages of technical triplicates. **i** Immunoblots of NUDT1 in 4-OHT induced SHEP MYCN-ER cells overexpressing NUDT1 WT or the E77K mutant, treated with LC-1-40 (50 nM) for 36 h. **j** Cell death analysis of 4-OHT induced SHEP MYCN-ER cells overexpressing NUDT1 WT or the E77K mutant, treated with LC-1-40 (50 nM). Data shown as averages of technical triplicates. Statistical significance was determined by one-way ANOVA (**h**) or two-way ANOVA (**j**). **k** Time course of plasma concentration of LC-1-40 in C57 mice (*n* = 3) after single IV (Intravenous), IP (Intraperitoneal), and PO (Oral) dose administration. Mean pharmacokinetics of LC-1-40 is shown on the right. Data are shown as mean ± SD. --, not calculated; $T_{1/2}$, terminal elimination phase half-life; $T_{max}$, time of the first occurrence of $C_{max}$; $C_{max}$, maximum observed concentration; $AUC_{inf}$, area under the concentration-time curve from time 0 to infinity; CL plasma clearance, *Vss* steady-state volume of distribution, F bioavailability. **c–f**, **h–j** these experiments were independently repeated three times with similar results. Source data are provided as a Source Data file.

damaged nucleotides in part due to MYC(N)/NOX4-augmented ROS production. Our results showed that NUDT1 is not a general target for cancer treatment. Instead, we found that *MYC(N)*-high tumor cells and tumor models in vivo were addicted to NUDT1 function. Such a synthetic lethality between MYC(N) hyperactivation and NUDT1 inhibition was further consolidated by the potent, on-target NUDT1 degrader, LC-1-40, we present. Given that a major hurdle in cancer treatment is the need to pinpoint the subset(s) of patients who might benefit from any emerging targeted therapy, we propose that MYC(N) hyperactivation might become a useful predictive biomarker in NUDT1-based treatment regimen. In conclusion, we demonstrate that the metabolic alterations induced by MYC(N) constitutes another source of unique metabolic rewiring that nominates NUDT1 depletion as a promising treatment strategy.

## Methods

### Ethics

All animal experiments were conducted in accordance with the ethical rules for animals and approved by the Institutional Animal Care and Use Committee of Wuhan University. For human primary neuroblastoma tumor samples obtained with informed consents from Children's Hospital of Fudan University, ethics approval was reviewed and granted by the Children's Hospital of Fudan University Institutional Review Board (No. 2019-224). After obtaining informed consent from guardians of each patient, the neuroblastoma patient samples were included in this study. All pathological specimens have been de-identified for the sake of patient privacy.

### Cell culture

Human neuroblastoma cell lines Kelly, BE-2C, NB-EBC1, SHEP, and SHEP MYCN-ER were kindly provided by Drs. John M Maris and Michael D. Hogarty (Children's Hospital of Philadelphia, University of Pennsylvania, Philadelphia, PA, USA). P493 cells were obtained from Dr Chi V Dang (Abramson Cancer Center, University of Pennsylvania, USA). SF188 cells were obtained from Dr Craig Thompson (Sloan-Kettering Cancer Center, USA). 293 T, A549, NCI-H82, SK-N-BE2, Ramos, and Daudi cells were purchased from American Type Culture Collection (ATCC). KOPTK1 cells were kindly provided by Dr Warren Pear (Abramson Cancer Center, University of Pennsylvania, USA). Human primary neuroblastoma tumor samples were obtained with informed consents from Children's Hospital of Fudan University with patient information listed in Supplementary Table 1. P493, SF188, Ramos, Daudi, NCI-H82, and neuroblastoma cell lines were cultured in RPMI-1640 (Hyclone) medium supplemented with 10% fetal bovine serum (FBS, Gibco) and 1% penicillin/streptomycin (Hyclone). KOPTK1 were grown in complete RPMI-1640 medium supplemented with 10% FBS and 1% penicillin/streptomycin, 1% non-essential amino acids (Gibco), 2 mM L-glutamine (Sigma–Aldrich), 1 mM sodium pyruvate (Sigma–Aldrich), and 55 μM β-mercaptoethanol (Sigma–Aldrich).

293 T and A549 cells were maintained in DMEM (Hyclone) containing 10% FBS and 1% penicillin /streptomycin. Patient-derived primary neuroblastoma cells were grown in DME/F12 medium (Sigma–Aldrich) containing 15% FBS, 1% non-essential amino acids, 2 mM L-glutamine, 1 mM sodium pyruvate, 1% B27 (Thermo Fisher), 10 ng/ml EGF (Sigma–Aldrich Aldrich), 15 ng/ml FGF (Peprotech) and 1% penicillin/streptomycin. All cells were cultured at 37 °C in a humidified incubator with 5% $CO_2$ and routinely tested for *Mycoplasma* contamination.

### Reagents and chemicals

Antibodies against β-Actin (AC026, 1:10000), NUDT1 (A13330, 1:1000 for immunoblots, 1:200 for immunoprecipitation), NOX4 (A11274, 1:1000), H3 (A2348, 1:1000), NCL (A5904, 1:1000) were obtained from ABclonal. PLK1 (4513 S, 1:1000), Phospho-PLK1 (Thr210) (5472 T, 1:1000), MYC (13987, 1:1000 for immunoblots), cleaved-Caspase 3 (Asp175) (9661 S, 1:1000 for immunoblots, 1:100 for IHC) were obtained from Cell Signaling technology. Flag-tag (F1804, 1 μg for immunoprecipitation) was obtained from Sigma–Aldrich. N-MYC (sc-53993, 1:1000 for immunoblots, 4 μg for ChIP) and PCNA (sc-56, 1:2000 for IHC) were obtained from Santa Cruz Biotechnology. 8-oxo-dG (AB5830, 1:300 for IHC) and γH2AX (05-636, 1:200 for IHC, 1:100 for IF, 1:1000 for immunoblots) were obtained from Millipore. HRP-conjugated goat anti-mouse (115-035-003, 1:5000) and anti-rabbit (111-035-003, 1:5000) secondary antibodies were obtained from Jackson ImmunoResearch Laboratories. The Phospho-NUDT1 S121 antibody was generated by Dia-An Technology by immunizing rabbits with phosphorylated S121 peptides (PDD-(phospho)S-YWF) conjugated with carrier protein keyhole limpet hemacyanin.

Other chemicals include 4-OH Tamoxifen, Tetracycline HCl, and TH287 (Selleck Chemicals); Acetylcysteine, BI6727, MG132, Pomalidomide and SBE-β-CD (MCE); CHX (Sigma–Aldrich); DAPI (Yeasen); Alexa488-conjugated avidin (Invitrogen).

### sgRNA library design and cloning

We constructed a metabolism-focused sgRNA library consisting of ~35,000 total sgRNAs targeting 2745 genes encoding human metabolic enzymes and transporters, along with 500 non-target control sequences (NTC), as previously described[33]. All sgRNA sequences are listed in Supplementary Data 2. The designed 20-nt target-specific sgRNA sequences were synthesized as a pool on microarray surfaces (Genscript). Overhangs compatible with Gibson Assembly were added to the sequences, which were inserted into the pSico-based barcoded sgLenti sgRNA library vector (Addgene) kindly provided by Dr Hao-peng Wang (School of Life Science and Technology, ShanghaiTech University, China). The synthesized sgRNA template sequence was as follows: 5′-AGTATCCCTTGGAGAACCACCT TGTTGG-(N)$_{20}$-GTTTAA-GAGCTATGCTGGAAACAGCATA-3′. Template pools were PCR-amplified using 2×Master Mix Phusion Flash (Thermo Fischer Scientific) according to the manufacturer's protocol with 20 ng oligo pool

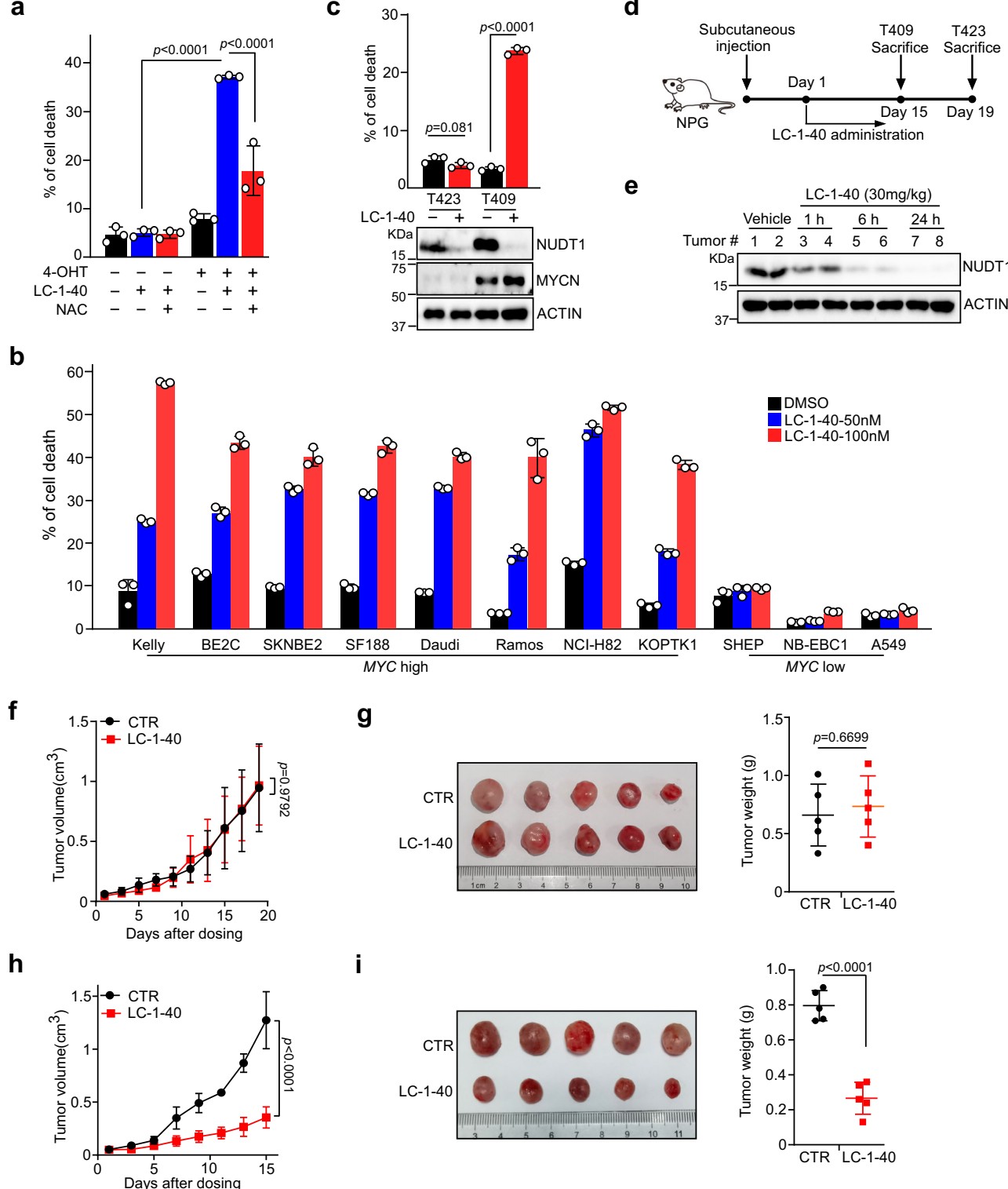

**Fig. 6 | LC-1–40 elicits MYC-driven synthetic lethality. a** Cell death analysis of SHEP MYCN-ER cells treated with LC-1-40 (50 nM) in the presence of 4-OHT (200 nM) and/or NAC (100 μM). **b** Cell death analysis of indicated tumor cells treated with LC-1-40. **c** Cell death analysis of primary neuroblastoma cells isolated from T423 (*MYCN* nonamplified) and T409 (*MYCN*-amplified) in the presence of LC-1-40 (100 nM). **d** Graphical illustration of neuroblastoma PDX and treatment strategy (see Materials and Methods for details). Mice were subjected to LC-1-40 (30 mg/kg) or vehicle treatment once daily. **e** Immunoblot of NUDT1 in T423 xenograft tumors isolated from NPG mice treated with vehicle or LC-1-40

(30 mg/kg, i.p.). Two tumors from each treatment group were used for the immunoblots. **f–i** Tumor growth and weights of T423 xenografts after 19-day treatment (**f**, **g**) or T409 xenografts after 15-day treatment (**h**, **i**) with vehicle or LC-1-40 (30 mg/kg). Five tumors were analyzed in each group. **a–c** data shown as averages of technical triplicates; **a–c**, **e** these experiments were independently repeated three times with similar results. Statistical significance was determined by unpaired two-tailed Student's *t*-test (**c**, **g**, **i**) or two-way ANOVA (**a**, **f**, **h**). Source data are provided as a Source Data file.

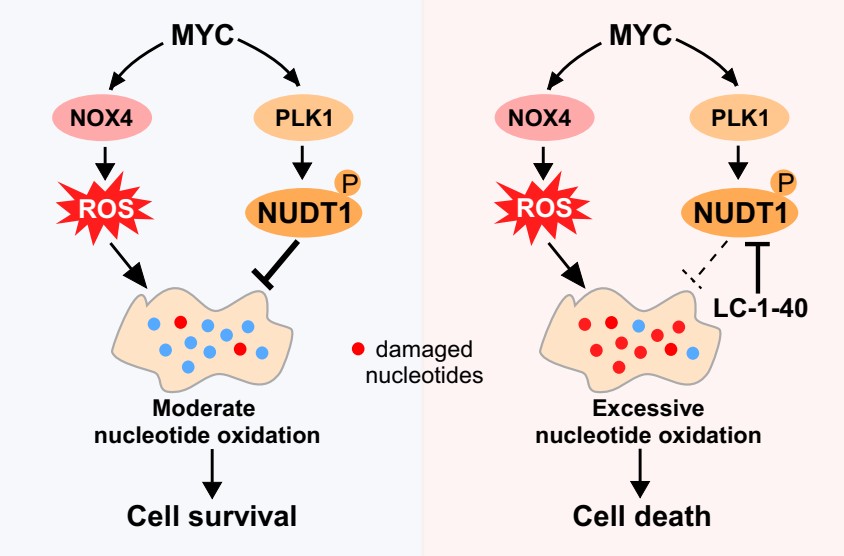

**Fig. 7 | Proposed mechanism for NUDT1-dependent survival in MYC-driven tumor cells.** MYC orchestrates the balance of two metabolic pathways that act in parallel, the NOX4-ROS pathway and the PLK1-NUDT1 nucleotide-sanitizing pathway. LC-1-40, which degrades NUDT1, disrupts the metabolic balance and elicits MYC-driven tumor cell death. See text for more details.

template, 100 μM forward primer (5′-GGAGAACCACCTTGTTGG-3′), 100 μM reverse primer (5′- GTTTCCAGCATAGCTCTTAAAC-3′) and the following PCR program: 1× (98 °C for 1 min), 19× (98 °C for 15 s, 55 °C for 15 s, 72 °C for 20 s), and 1× (72 °C for 5 min, 12 °C forever). PCR products were purified by Minelute columns (Qiagen). The sgLenti vector was digested by AarI (Thermo Fischer Scientific) at 37 °C overnight, followed by excision from agarose gel and purification by the Nucleo-Spin gel extraction kit (Macherey-Nagel). 330 ng digested sgLenti and 50 ng amplified oligo library were diluted in a total 40 μL volume and subjected to ligation according to the instructions of ClonExpress II One Step Cloning Kit (Vazyme). The resulting products were precipitated by isopropanol and transformed into Endura-competent cells (Lucigen). The library cloning coverage (number of E. coli colonies per sgRNA plasmid) was determined to be >100×.

### sgRNA screening
Cas9-expressing SHEP MYCN-ER cells were transduced with the pooled lentiviral library at a multiplicity of infection (MOI) of 0.3 to ensure that only one gene was targeted for Cas9-mediated editing in each cell. mCherry-positive infected cells were selected by flow cytometry and subjected to sgRNA screening. Approximately 35 million cells were first harvested for genomic DNA extraction, and the remaining cells were continuously treated with ethanol or 4-OHT (50 nM) and cultured for 14 generations before harvesting for genomic DNA extraction. Cells were counted as a routine throughout the screening to ensure equal population doublings of ethanol and 4-OHT treated cells. Genomic DNA was extracted using DNAzol (Invitrogen) and precipitated with ethanol. Multiple PCR reactions were conducted to allow amplification of genomic DNA from a 1000× cell coverage for each sample. The resulting PCR product (344 bp) was extracted from agarose gel, which were subjected to sequencing on an Illumina HiSEq and analyzed by the MAGeCK program.

### Cell death assay
Cell death was analyzed using the Annexin V-FITC Apoptosis Kit (Biovision). $1 \times 10^5$ cells were resuspended in 100 μl binding buffer with 1 μl Annexin V-FITC and 2 μl PI for 10 min at room temperature. Acquisition was performed on the Accuri C6 (BD Biosciences) and data were analyzed with FlowJo software (Tree Star).

### Immunoblot and immunoprecipitation
For immunoblot, whole-cell extracts were prepared using RIPA buffer[34]. Lysates were cleared by centrifugation and protein concentrations were determined by BCA Assay Kit (Thermo Fisher Scientific). 30–50 μg total proteins were subjected to SDS-PAGE and transferred to PVDF membrane (Bio-Rad). Blots were blocked by 5% fat free milk for 1 h at room temperature and then incubated with primary antibodies at 4 °C overnight. After washing, appropriate secondary antibodies were applied at room temperature before detection with Super Signal Chemiluminescent Substrate (Bio-Rad).

For immunoprecipitation, cells were lysed in IP buffer[34], centrifuged at $12,000 \times g$ for 10 min at 4 °C. The resulting supernatants were pre-cleared with protein G agarose beads (GE) for 1 h at 4 °C. Antibodies were used to precipitate proteins overnight at 4 °C, followed by incubation with protein G agarose beads. Immunoprecipitated proteins were washed with IP buffer three times and then eluted with loading buffer prior to SDS-PAGE and immunoblotting.

### Nucleotide pool analysis
Cellular nucleotides were extracted from SHEP MYCN-ER cells using 0.4 N perchloric acid and neutralized with potassium chloride. Ribonucleotides were selectively eliminated through a boronate affinity column, while deoxynucleotides were subjected to HPLC analysis. Identification and quantitation were performed using UV absorbance at 254 and 281 nm as previously described[35].

### 8-oxo-dG assay
Avidin binds to 8-oxo-dG with high specificity[36] and was therefore used for 8-oxo-dG measurement. Cells were fixed on coverslip by methanol at −20 °C for 20 min and permeabilized with 0.1% Triton X-100 for 15 min at room temperature, before blocked by 5% bovine serum albumin, 0.1% Triton X-100 in TBS for 2 h at room temperature. Cells were then incubated with 10 μg/ml Alexa488-conjugated avidin (Invitrogen) in blocking solution for 2 h, then rinsed three times in TBS with 0.1% Triton X-100. DNA was finally counterstained with DAPI and slides were mounted and analyzed by confocal microscopy (Carl Zeiss).

## Quantitative RT-PCR

Total RNA was isolated using Trizol reagent. 1 µg RNA was reversed transcribed using the ReverTra Ace qPCR RT kit (TOYOBO). Quantitative PCR was carried out on a CFX Connect Real-Time PCR System (Bio-Rad) using the ChamQ SYBR qPCR Master Mix (Vazyme). Primer sequences for qPCR are listed in Supplementary Table 2.

## Chromatin immunoprecipitation (ChIP)

SHEP MYCN-ER cells were fixed with 1% formaldehyde diluted in PBS (Phosphate buffer saline) for 10 min at room temperature, quenched with 0.125 M glycine for 5 min at 37 °C. Fixed cells were lysed with SDS Lysis Buffer, followed by sonication (Bioruptor Pico Sonifier, Diagenode) to shear chromatin DNA to a size range of 500–1000 bp. Precleared chromatin was immunoprecipitated with antibody against N-MYC overnight at 4 °C. Antibody-chromatin complexes were pulled down by pre-blocked Protein A/G beads (Smart-Lifesciences) for 1 h at 4 °C. De-crosslinked DNA was subjected to qPCR analysis using specific primers listed in Supplementary Table 2.

## Expression and purification of NUDT1 and NUDT1 mutants

GST-tagged NUDT1 and mutants were expressed from the pGEX-4T1 vector in *E. coli* BL21 (DE3) strain upon 100 µM IPTG induction for 5 h. Bacteria were resuspended in binding buffer (140 mM NaCl, 2.7 mM KCl, 10 mM $Na_2HPO4$, 1.8 mM $KH_2PO4$, 0.1% Tween20, pH 7.4) prior to sonication and centrifugation. The resulting supernatants were subjected to purification through Glutathione Agarose beads (Cube biotech). GST-tagged protein-coupled beads were washed with binding buffer 5–10 times and then eluted by 20 mM Glutathione solution.

## In vitro protein interaction assay

GST-NUDT1 or GST control protein bound to Glutathione Agarose beads was incubated with His-PLK1 protein (Sino Biological Incorporation) in IP buffer overnight at 4 °C. Beads were then washed thrice with IP buffer, the associated proteins were eluted in SDS-PAGE loading buffer, followed by Coomassie staining and immunoblot analysis.

## In vitro protein kinase assay

PLK1 kinase active (PLK1-T210D) or dead (PLK1-K82R) mutants were expressed in 293 T cells and purified by immunoprecipitation using PLK1 antibody coupled to protein G agarose beads. GST-NUDT1 and its mutants were expressed in *E. coli* BL21(DE3) strain. Kinase and substrates (2 µg GST fusion proteins) were incubated in kinase buffer (Cell Signaling Technologies) with the presence of 5 µCi [γ-$^{32}$P] ATP (PerkinElmer) and 200 µM cold ATP (Thermo Fisher Scientific) for 1 h at 30 °C. Reactions were stopped by addition of SDS sample buffer, and samples were then heated for 5 min at 95 °C before analysis by SDS-PAGE and autoradiography.

## Measurement of 8-oxo-dGTPase activities

NUDT1 enzyme activity assay was performed as described with minor modifications[20,29,37]. In Fig. 3a, endogenous NUDT1 immunoprecipitated with NUDT1 antibody coupled to Protein G beads, and on-beads catalytic activity was assayed in reaction buffer (0.1 M Tris-acetate, pH 7.5, 40 mM NaCl, 10 mM magnesium acetate, 2 mM dithiothreitol, 0.005% Tween20). After addition of the mixed reagent from the PPi-Light Inorganic Pyrophosphate Assay kit (Lonza), 8-oxo-dGTP (13.2 µM; TriLink BioTechnologies) was added as a substrate to initiate the enzymatic reaction. Reaction mixtures in a total volume of 100 µl were incubated on a plate shaker at room temperature for 10 min. Relative NUDT1 activity was measured by monitoring the pyrophosphate (PPi) generated through NUDT1-catalyzed nucleotide triphosphate hydrolysis. The bioluminescent signal was recorded for 30 min using MD SpectraMax i3x (Molecular Devices). In Fig. 3g, exogenous Flag-NUDT1 in 293 T cells were purified and on-beads catalytic activity was assayed as described.

To assess NUDT1 (WT or mutant) enzyme activity, kinetic parameters of NUDT1 or its mutants with 8-oxo-dGTP were determined by assaying initial velocities at substrate concentrations ranging from 0 to 50 µM in reaction buffer (volume 40 µl). 0.5 nM purified protein was included, and the reaction mixtures were incubated on a plate shaker at room temperature for 10 min. The resulting PPi was detected by the Light Inorganic Pyrophosphatase Assay Kit, and standard curve was used to convert bioluminescent signal to PPi concentration. The $kcat$, $K_M$, and $kcat/K_M$ values were calculated by the Michaelis–Menten equation and non-linear regression analysis in the GraphPad Prism 9.

## Mass spectrometry analysis

Total protein extraction from SHEP MYCN-ER cells stably expressing Flag-tagged NUDT1 was subjected to immunoprecipitation using Flag antibody. Immunoprecipitated proteins were washed with IP buffer three times and then eluted with 0.2 M glycine pH 2.0. Eluted proteins were precipitated with four volumes of acetone and then re-solubilized in urea prior to trypsin (Promega) digestion at 37 °C for 20 h. Digested peptides were desalted using the C18 Stage tips (Thermo Fisher Scientific) and loaded onto an EASY-nLC system for LC–MS/MS analysis by Orbitrap Exploris 480 mass spectrometer equipped with the FAIMS Pro interface (Thermo Fisher Scientific). Mass spectrometry resolutions were set to 60,000 at m/z 200 and mass range was set to 350–1500. Raw data were processed with Proteome Discoverer 2.4 (Thermo Fisher Scientific). Criteria to screen specific NUDT1-interacting proteins in 4-OHT treated group: $p$-value < 0.05 with a fold change >2. The list is shown in Supplementary Data 3.

## Mass spectrometry analysis of phosphopeptides

PLK1 kinase active mutant T210D was incubated with GST-NUDT1 (2 µg) in kinase buffer for 1 h at 30 °C. The reaction mix was subjected to SDS-PAGE and Coomassie blue staining. Gel bands corresponding to GST-NUDT1 were excised and extracted proteins were digested with trypsin in 50 mM ammonium bicarbonate at 37 °C overnight. After treatment with 5 mM dithiothreitol and 11 mM iodoacetamide, the resulting peptides were separated by a silica capillary column and eluted at a flow rate of 0.3 mL/min using the UltiMate 3000 HPLC system (Thermo Fisher Scientific), coupled with the Q Exactive mass spectrometer (Thermo Fisher Scientific). Data-dependent acquisition mode was adjusted by the Xcalibur 2.2 software. LC–MS/MS analysis was performed in the Protein Chemistry Facility, Center of Biomedical Analysis at Tsinghua University (Beijing, China) according to previous reports[38].

## Synthetic procedures of LC-1–40

Unless otherwise indicated, starting materials, reagents, and solvents were used without further purification as received from commercial suppliers. All reactions carried out were monitored using a Waters Acquity UPLC/MS system with a mass range of 100–1150. Purification of reaction products were performed by flash column chromatography using Santai SepaBean™ machine system with SepaFlash™ columns (4, 12, 24, 40, and 80 g) and a Waters prep-HPLC system on a Waters Sunfire C18 column (19 mm × 50 mm, 5 µM): solvent gradient = 95% A at 0 min, 15% A at 22 min, and 5% A from 22 min to 28 min; solvent A = 0.035% trifluoroacetic acid (TFA) in Water; Solvent B = 0.035% TFA in MeOH; flow rate: 20 mL/min. $^1$HNMR spectra were recorded on 400 MHz Bruker Avance III spectrometer, and chemical shifts are presented in parts per million (ppm, δ) downfield from tetramethylsilane (TMS). Chemical shifts are reported relative to deuterated dimethyl sulfoxide (DMSO) (δ = 2.50) for $^1$HNMR. Spectra are given in ppm (δ) and spin multiplicities are described as br = broad, s = singlet, d = doublet, t = triplet, q = quartet, and m = multiplet. Coupling constants (J) are reported in Hz.

Synthesis of -(2,6-dioxopiperidin-3-yl)-5-(4-(piperidin-4-ylmethyl) piperazin-1-yl)isoindoline-1,3-dione (5).

To a solution of compound 1 (2.76 g, 10 mmol) and *tert*-butyl piperazine-1-carboxylate (1.86 g, 10 mmol) in 50 mL of DMSO was added *N,N*-Diisopropylethylamine (DIEA) (8.7 mL, 50 mmol). The mixture was stirred at 90 °C for 12 h. The reaction mixture was diluted with 150 mL of $H_2O$ and extracted with 300 mL of ethyl acetate (EtOAc). The organic phase was washed with brine (150 mL × 2), dried over anhydrous $Na_2SO_4$, filtered, and concentrated under reduced pressure to give a residue, which was then purified by flash column chromatography to give compound 2 as a yellow solid (4.17 g, 94.3%). LC−MS: *m/z* 465.3 [M+Na]. ¹HNMR: 400 MHz, DMSO-$d_6$ δ 11.08 (s, 1H),

2.06−1.97 (m, 1H), 1.69 (br d, *J* = 11.2 Hz, 3H), 1.39-1.38 (m, 9H), 1.02-0.90 (m, 2H).

To a solution of compound 4 (3.75 g, 6.95 mmol) in 100 mL of DCM was added 30 mL of TFA. The mixture was stirred at 25 °C for 2 h. The reaction mixture was then evaporated to give a crude product 5, which was directly used for the next step without further purification. LC−MS: *m/z* 440.3 [M + 1].

Synthesis of *N*-cyclopropyl-6-(4-(4-((4-(2-(2,6-dioxopiperidin-3-yl)-1,3-dioxoisoindolin-5-yl)piperazin-1-yl)methyl)piperidine-1-carbonyl)phenyl)-4-(phenylamino)quinoline-3-carboxamide (LC-1-40).

**LC-1-40**

7.69 (d, *J* = 8.4 Hz, 1H), 7.34 (d, *J* = 2.0 Hz, 1H), 7.24 (dd, *J* = 2.4, 8.8 Hz, 1H), 5.07 (dd, *J* = 5.2, 12.8 Hz, 1H), 3.47 (s, 8H) 2.95−2.81 (m, 1H), 2.64−2.53 (m, 2H), 2.07−1.99 (m, 1H), 1.42 (s, 9H).

To a solution of compound 2 (4.17 g, 9.4 mmol) in 150 mL of DCM was added 50 mL of TFA. The mixture was stirred at 25 °C for 5 h. The reaction mixture was then evaporated to give a crude product 3, which was directly used for the next step without further purification. LC−MS: *m/z* 343.2 [M + 1].

To a solution of compound 3 (4.17 g, 9.4 mmol) and *tert*-butyl 4-formylpiperidine-1-carboxylate (2.0 g, 9.4 mmol) in 100 mL of DMF was added NaBH(OAc)$_3$ (3.9 g, 18.8 mmol) slowly. The mixture was stirred at 25 °C for 12 h. The reaction mixture was then diluted with 300 mL of $H_2O$ and extracted with 500 mL of EtOAc. The organic phase was washed with brine (250 mL × 2), dried over anhydrous $Na_2SO_4$, filtered, and concentrated under reduced pressure. The crude product was purified by reverse phase HPLC (5−95% MeOH in $H_2O$) to give compound 4 as a yellow solid (3.75 g, 74%). LC−MS: *m/z* 540.4 [M + 1]. ¹HNMR: 400 MHz, DMSO-$d_6$ δ 11.08 (s, 1H), 7.68 (br d, *J* = 8.8 Hz, 1H), 7.33 (br s, 1H), 7.25 (br d, *J* = 8.0 Hz, 1H), 5.07 (dd, *J* = 5.2, 12.8 Hz, 1H), 3.92 (br d, *J* = 11.6 Hz, 2H), 3.42 (br s, 4H), 2.93−2.83 (m, 2H), 2.73 (s, 2H), 2.63−2.53 (m, 2H), 2.47 (br d, *J* = 3.2 Hz, 3H), 2.17 (br d, *J* = 6.0 Hz, 2H),

To a solution of compound 6 (1.46 g, 5.4 mmol, 1.0 eq), aniline (0.59 mL, 6.48 mmol) in 30 mL of NMP was added AcOH (0.31 mL, 5.4 mmol). The mixture was stirred at 100 °C for 4 h. When compound 6 was consumed completely monitored by LC−MS, the mixture was cooled to room temperature, and the pH of reaction mixture was basified to 7−8 with saturated NaHCO$_3$, and the desired product precipitated. The suspension was then filtered and the solid compound 7 collected was directly used for the next step without further purification (1.2 g, 68%). LC−MS: *m/z* 327.2 [M + 1]. ¹HNMR: 400 MHz, DMSO-$d_6$ δ 9.70 (s, 1H), 8.88 (s, 1H), 8.16 (d, *J* = 2.4 Hz, 1H), 7.95 (d, *J* = 9.0 Hz, 1H), 7.80−7.76 (m, 1H), 7.37−7.28 (m, 2H), 7.13−7.03 (m, 3H), 3.92 (q, *J* = 7.2 Hz, 2H), 1.11 (t, *J* = 7.2 Hz, 3H).

To a solution of compound 7 (1.2 g, 3.67 mmol) in 10 mL of THF and 5 mL of $H_2O$ was added LiOH•$H_2O$ (770 mg, 18.35 mmol). When compound 7 was consumed monitored by LC−MS, the reaction mixture was extracted with 150 mL of CHCl$_3$ and isopropanol (v:v = 4:1). The organic layer was washed with 50 mL of 0.50 M HCl, 50 mL of saturated NaHCO$_3$, and 50 mL of brine, dried over anhydrous $Na_2SO_4$, and evaporated to give the desired product 8 as a yellow solid (0.97 g, 88%). LC−MS: *m/z* 299.2[M + 1]. ¹HNMR: 400 MHz, DMSO-$d_6$ δ 8.99 (s, 1H), 7.86 (br d, *J* = 9.0 Hz, 1H), 7.68 (dd, *J* = 2.0, 6.8 Hz, 1H), 7.55 (br s,

1H), 7.30 (br t, $J = 7.6$ Hz, 2H), 7.13 (br t, $J = 7.4$ Hz, 1H), 7.07 (br d, $J = 7.8$ Hz, 2H).

To a solution of compound 8 (0.97 g, 3.23 mmol) and cyclopropanamine (368 mg, 6.46 mmol) in 30 mL of DMF, HATU (1.8 g, 4.85 mmol) and DIEA (2.8 mL, 16.15 mmol) were added and the mixture was stirred at room temperature for 1 h. When compound 8 was consumed completely, the reaction mixture was diluted with 300 mL EtOAc, washed with brine (120 mL × 3), dried over anhydrous $Na_2SO_4$, filtered, and concentrated under reduced pressure. The residue was then purified by column chromatography to give compound 9 as a yellow solid (0.99 g, 91%). LC–MS: $m/z$ 338.2[M + 1]. [1]HNMR: 400 MHz, DMSO-$d_6$ δ 9.41 (s, 1H), 8.50 (s, 1H), 8.26 (d, $J = 1.6$ Hz, 1H), 8.01 (d, $J = 1.0$ Hz, 1H), 7.77–7.69 (m, 2H), 7.53 (dd, $J = 2.2, 5.6$ Hz, 1H), 7.09–6.99 (m, 2H), 6.80 (t, $J = 7.4$ Hz, 1H), 6.75 (d, $J = 3.8$ Hz, 2H), 0.28-0.25 (m, 2H).

To a solution of compound 9 (0.99 g, 2.9 mmol) and (4-(methoxycarbonyl)phenyl)boronic acid (522 mg, 2.9 mmol) in 30 mL of THF and 10 mL of $H_2O$ was added $K_3PO_4$ (1.5 g, 5.8 mmol) and Xphos-Pd-$G_2$ (23 mg, 0.029 mmol). The mixture was stirred at 80 °C for 2 h. The reaction mixture was diluted with 30 mL of $H_2O$ and extracted with EtOAc (100 mL × 3). The combined organic layers were washed with brine (100 mL × 2), dried over $Na_2SO_4$, filtered, and concentrated under reduced pressure to give a residue. The residue was then purified by column chromatography to give compound 10 as a yellow solid (0.9 g, 71%). LC–MS: $m/z$ 438.2[M + 1].

To a solution of compound 10 (0.87 g, 2 mmol) in 10 mL of THF and 10 mL of MeOH was added 0.5 mL of 1.0 N aqueous LiOH. The mixture was stirred at 25 °C for 5 h. The reaction mixture was diluted with 20 mL of $H_2O$ and extracted with 100 mL of EtOAc (50 mL × 2). The combined organic layers were washed with 10 mL of 0.5 N HCl, 30 mL of saturated $NaHCO_3$, and 30 mL of brine, dried over $Na_2SO_4$, filtered, and concentrated under reduced pressure. The crude product 11 was directly used for the next step without further purification (0.78 g, 93%). LC–MS: $m/z$ 424.2[M + 1].

To a solution of compound 11 (0.78 g, 1.86 mmol) and compound 5 (0.818 g, 1.86 mmol) in 20 mL of DMF was added HATU (1 g, 2.79 mmol) and DIEA (1.6 mL, 9.3 mmol). The mixture was stirred at 25 °C for 1 h. The reaction mixture was then diluted with 60 mL of $H_2O$ and extracted with DCM. The organic layer was dried over anhydrous $Na_2SO_4$, filtered, and concentrated under reduced pressure. The crude product was purified by reverse phase HPLC (5–95% MeOH in $H_2O$) to give compound LC-1-40 as a yellow solid (1.3 g, 83%). LC–MS: $m/z$ 845.7[M + 1]. [1]HNMR: 400 MHz, DMSO-$d_6$ δ 10.74 (br s, 1H), 8.79 (s, 1H), 8.58–8.27 (m, 1H), 8.03 (d, $J = 9.2$ Hz, 1H), 7.90–7.80 (m, 2H), 7.69 (d, $J = 8.4$ Hz, 1H), 7.36–7.30 (m, 4H), 7.28 (br d, $J = 2.0$ Hz, 1H), 7.23–7.14 (m, 3H), 7.10–7.03 (m, 3H), 6.88 (br s, 1H), 4.95 (dd, $J = 5.2, 12.4$ Hz, 1H), 4.79–4.63 (m, 1H), 3.96–3.64 (m, 1H), 3.41 (br d, $J = 4.8$ Hz, 4H), 3.07–2.72 (m, 6H), 2.57 (br s, 4H), 2.28 (br d, $J = 6.4$ Hz, 2H), 2.17–2.09 (m, 1H), 1.97–1.77 (m, 4H), 1.40–1.02 (m, 3H), 0.96–0.85 (m, 2H), 0.74–0.66 (m, 2H).

## TMT-labeled quantitative proteomics assay

SF188 cells were treated with DMSO or 100 nM LC-1-40 in biological duplicates for 6 h and harvested by centrifugation at 4 °C, 1000 g. Cells were then washed twice with cold PBS, and lysed in buffer (20 mM HEPES, 300 mM NaCl, 1 M Urea, 1% NP-40 Substitute, 10 mM $MgCl_2$, 1 mM DTT) on ice for 30 min, followed by centrifugation at 12,000 g for 15 min. Collected supernatant was quantified by BCA kit, and 100 μg of protein from each sample were used for digestion and TMT labeling. Quantitative proteomics were performed using Deng's method[39]. Equal amounts of protein were reduced with 5 mM dithiothreitol and alkylated with 13 mM iodoacetamide. Proteins were digested overnight at 37 °C by trypsin overnight, and the resulting peptides were desalted using C18 stage tips, dried by a speedvac (Thermo Fischer Scientific) and reconstituted in 100 mM triethyl ammonium bicarbonate (TEAB). Tandem mass tag (TMT) label reagents (Thermo Fischer Scientific)

dissolved in acetonitrile were added to the peptide solutions and incubated at room temperature for 1 h. The reaction was quenched by the addition of 5% hydroxylamine for 15 min. Desalted samples were separated by EASY-nLC System coupled to an Orbitrap Exploris 480 mass spectrometer. Collected signals were analyzed by Proteome Discovery v.2.4.

## Animal studies

All mice were maintained in Specific Pathogen Free (SPF) animal facility of Medical Research Institute, Wuhan University. Mice were housed in groups of 4-6 mice in an individually ventilated cage (IVC) in a 12:12 light-dark cycle (08:30–20:30 light; 20:30–8:30 dark). The ambient temperature was 22 ± 2 °C with 50–60% relative humidity. Transgenic TH-*MYCN* mice in 129 S/v background were kindly provided by Dr William A Weiss (University of California, San Francisco). *Nudt1*$^{-/-}$ mice in C57BL6 background were generated by the efficient CRISPR/Cas9-mediated genome editing which depletes 4 kb in Nudt1 locus (Biocytogen). To obtain N-MYC driven mouse neuroblastoma model, *Nudt1*$^{-/-}$ mice were backcrossed twice to TH-*MYCN*$^{+/-}$ mice and generated a mice strain in mixed background (129/B6). TH-*MYCN*$^{+/+}$*Nudt1*$^{+/+}$ and TH-*MYCN*$^{+/+}$*Nudt1*$^{-/-}$ mice in 129/B6 background were generated to monitor the incidence of tumor onset and animal survival. Genotyping was performed using gene-specific PCR primers listed in Supplementary Table 2.

For NOTCH1(ICN1)-induced leukemia model, bone marrow cells isolated from 8-week-old donor mice (2 hind legs) were treated with magnetic beads (Lineage Cell Depletion Kit, Miltenyi Biotec) to obtain lineage-negative (Lin$^-$) hematopoietic progenitor cells. Enriched cells were collected and incubated for 24 h in the presence of cytokine mix: 20 ng/mL mFLT3-L, 20 ng/mL mTPO, and 100 ng/mL mSCF (Pepro-Tech). Progenitor cells were transduced two rounds with MigR1-ICN1 retroviruses (1000 g, 90 min, 32 °C) along with 6 μg/mL polybrene. One million cells were tail intravenously injected into sub-lethally (5.5 Gy) irradiated 8-week-old C57BL/6 female recipient mice, which were then treated with enrofloxacin-containing water (Baytril, Bayer) for 2 weeks. The percentage of GFP$^+$ cells in peripheral blood was analyzed by flow cytometry to trace the leukemia initiation. Mice were monitored for survival and euthanized when moribund or demonstrating obvious clinical distress.

For patient-derived xenografts, human primary neuroblastoma tumors were transplanted into 6-week-old NPG female mice (Beijing Vitalstar Biotechnology) at least two passages prior to the experiment. Tumor slices with an average volume of 5 $mm^3$ were subcutaneously inoculated into 6-week-old recipients. Tumor volumes were measured by calipers every other day and calculated as (length × width$^2$ /2). Mice were randomized divided into two treatment groups (vehicle and LC-1-40) when established tumors reached ~50 $mm^3$. LC-1-40 was dissolved in 10% DMSO, 90% saline (with 20% SBE-β-CD). Mice were treated with 30 mg/kg LC-1-40 (i.p.; daily) or vehicle control. Upon treatment termination, mice were sacrificed with cervical dislocation, and subcutaneous tumors were resected for tumor weight assessment. During the animal experiments, the maximal tumor burden permitted by the Institutional Animal Care and Use Committee of Wuhan University is 1500 $mm^3$. Thus, when tumor volumes reached a maximum of 1500 $mm^3$ or the tumor had ulcers with diameter reached 1 cm, the mice were immediately euthanized.

## Immunohistochemistry (IHC)

The IHC analysis was carried out using Histostain-Plus IHC Kit (Thermo Fisher Scientific). Tumor samples were embedded in paraffin and sectioned at 6 μm. Sections were incubated with the antibodies against 8-oxo-dG, γH2AX or c-Caspase 3 overnight at 4 °C, then subjected to horseradish peroxidase-linked secondary antibodies for 1 h at room temperature. Staining was visualized by DAB staining (Vector Laboratories). Representative stains were imaged

at × 400 magnification by Digital Pathology Scanner and quantified using ImageJ software.

## Quantification and statistical analysis

Statistical analysis was carried out using GraphPad Prism 9. Comparisons of two groups were analyzed using unpaired two-tailed Student's t-test, and statistical significance from three or more groups were calculated by one-way or two-way ANOVA. Survival in mouse experiments was presented as Kaplan–Meier curves and significance was estimated by log-rank test. Differences were considered significant when $p < 0.05$.

## Reporting summary

Further information on research design is available in the Nature Portfolio Reporting Summary linked to this article.

## Data availability

The sgRNA screening data generated in this study have been deposited in the NCBI database under accession code PRJNA989594. The mass spectrometry proteomics data and TMT-labeled quantitative proteomics data generated in this study have been deposited in the ProteomeXchange database under accession code PXD043444. The remaining data are available within the Article, Supplementary Information or Source Data file. Source data are provided with this paper.

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

## Acknowledgements

We thank members of the Qing, Jiang, and Liu laboratories for helpful suggestions. We thank the Core Facility of Medical Research Institute at Wuhan University for technical support and the Research Center for Medicine and Structural Biology of Wuhan University for proteomic assistance. This study was supported by grants from National Key R&D Program of China (2022YFA1103200 and 2021YFA1100501), National Natural Science Foundation of China (82230092 and 81830084 to G.Q. and 82161138024 to H.L.), Hubei Provincial Natural Science Fund for Creative Research Groups (2021CFA003 to H.L.), Fundamental Research Funds for the Central Universities (2042022dx0003), and Wuhan University start-up funds (B.J.).

## Author contributions

G.Q. conceived the project and designed the study. G.Q. and B.J. supervised the study. G.Q., H.L. and M.Y. wrote the manuscript. M.Y. and Y.F. performed most of the experiments. M.Y., Y.F. and F.W. performed sgRNA screening. M.Y. and M.H. performed mass spectrum analysis of phosphopeptides. M.Y., Z.S. and R.X. performed mass spectrum analysis and TMT-labeled quantitative proteomics. B.J. and L.C. designed and synthesized the PROTAC. M.Y., Y.F., Q.B., H.H. and J.X. conducted animal experiments. Z.W. and S.G. carried out the structure analysis. All authors contributed to the final manuscript.

## Competing interests

M.Y., Y.F., L.C., B.J. and G.Q. are inventors on a patent application in China related to therapeutic targeting of NUDT1 in treatment of MYC(N)-driven cancers (202311783418.X). The key findings regarding synthetic lethality between NUDT1 inhibition and MYC(N) activation as well as NUDT1-targeting molecules disclosed in this manuscript are a subject of patent application filed by Wuhan University. All remaining authors declare no competing interests.
