## [Peer Review File · Nature Communications]

Point-by-point response to reviewer comments:

We would like to express our gratitude to the reviewers for their thorough and insightful comments on our manuscript. Addressing each comment has strengthened our study and clarified our manuscript. Our point-by-point responses are below, and updates in manuscript are in red text for ease of review.

Reviewer #1 (Remarks to the Author):

Here, Ye et al. investigated the metabolic dependencies induced by MYC activation in neuroblastoma. To this end, the authors first performed a CRISPR screen with a targeted library focusing on metabolic enzymes in a MYC-ER system in a neuroblastoma cell line. They identified a hydrolase, NUDT1, as a TOP hit; NUDT1 prevents the incorporation of oxidized sGTP into DNA and thus reduces genotoxic stress. They validated this dependency in several MYC- and MYCN-driven cancer cell lines. To find out why MYCN induces the dependence of cancer cells on NUDT1 and leads to increased ROS production, they analyzed gene expression changes after 24 hours of MYCN-ER activation. They showed that the NOX4 gene is strongly induced by MYCN and that depletion of NOX4 partially prevents MYCN-mediated increased ROS production. Interestingly, MYCN also induces NUDT1 activity, but not through increased transcription, but indirectly through activation of PLK. PLK-mediated phosphorylation of NUDT1 activates the enzymatic activity of NUDT1. Next, they investigated whether KO of NUDT1 is tumor protective in a neuroblastoma model. This is the case; however, most NUDT1 KO animals also develop neuroblastoma with a delay and reach the experimental endpoint. Similar observations have been made in a T-ALL model.

Finally, the development of metabolically stable NUDT1 PROTACs (based on previously published PROTACS) is described. The lead compound, LC-1-40, slows tumor growth in a xenograft setting but does not induce tumor regression.

This is a comprehensive and multidisciplinary study. The observations are interesting from a mechanistic perspective and may provide an intriguing therapeutic approach. However, the effects of PROTACs in mice are modest. Overall, I consider this manuscript a strong candidate for Nature Communications if the following points can be addressed:

We appreciate the Reviewer's expert comment.

1. A pathway analysis of the differential hits (OHT, up and down) of the CRISPR screen would be interesting.

Based on the Reviewer's suggestion, we performed a pathway analysis of the gene sets with a significant \log_2 fold change ($p < 0.05$) (revised Supplementary Fig. 1f, g). Gene ontology (GO) analysis suggest that inhibition of genes related to DNA protection and glutathione metabolic process promotes MYC-driven tumor cell death, whereas depletion of those involved in protein glycosylation and ceramide biosynthesis promotes cell survival.

2. The experiments defining NOX4 as a direct MYC target are not convincing. The promoters of all expressed genes are bound to MYC. It is important to show whether the binding changes under MYCN activation (OHT) at the promoter and whether the induction of NOX4 can be seen even a few hours after OHT addition (and not only after 24 h).

Based on the Reviewer's comment, we pretreated SHEP MYCN-ER cells with 10 µg/ml cycloheximide (CHX) to shut down the global protein translation before a time-course induction of MYCN by 4-OHT. As expected, chromatin immunoprecipitation assays revealed that MYCN was selectively recruited to the E box region of the NOX4 gene upon 4-OHT induction in a time-dependent manner, proportionally leading to increased NOX4 mRNA levels (revised Fig. 2c and Supplementary Fig. 2f), supporting that MYCN directly activated NOX4 transcription to elevate ROS production. We agree with the Reviewer that both MYC and MYCN can bind the promoters of all expressed genes. Most likely, MYC(N) amplifies the NOX4 transcription at later time points.

3. It should also be investigated whether the MYC-induced overexpression of NOX4 is necessary and sufficient for the increased ROS production. For example, Figure 2d lacks the effect of NOX4 depletion in cells not treated with OHT. Also, NOX4 should be overexpressed in the system to show if NOX4 rate is limiting. It would also be interesting and important to analyze whether overexpression of NOX4 makes the cells sensitive to KO of NUDT1.

We appreciate the Reviewer's excellent suggestion. We repeated the experiment to assess the impact of NOX4 depletion on ROS production in SHEP MYCN-ER cells without 4-OHT treatment. Interestingly, NOX4 depletion exhibited a minimal effect on ROS levels (revised Fig. 2d), suggesting that it is not critical for basal ROS production when MYCN is not hyperactivated. As expected, NOX4 overexpression in SHEP MYCN-ER cells further increased ROS production (revised Supplementary Fig. 2j), supporting that NOX4 plays an important role in MYCN augmentation of ROS generation.

We then respectively overexpressed NOX4 in SHEP (MYC-low), SF188 (MYC-amplified) and Kelly (MYCN-amplified) cells, and found that NOX4 overexpression significantly enhanced SF188 and Kelly cell death upon NUDT1 depletion while this death-inducing effect was not observed in SHEP cells (revised Supplementary Fig. 2m), supporting that NUDT1 inhibition constitutes a synthetic lethal interaction with MYC(N) activation.

4. It is not clear to me why a PROTAC approach was chosen since overexpression of catalytically inactive NUDT1 did not rescue the NUDT1 gene knockout. Are catalytic inhibitors therapeutic?

The Reviewer raised an interesting scientific question. Indeed, overexpression of the catalytically inactive NUDT1 mutant can hardly rescue NUDT1-depletion induced cell death, supporting that the catalytic activity of NUDT1 is required for tumor cells to adapt to MYC(N)-driven metabolic reprogramming. In principle, catalytic inhibitors could be used as a pharmacological approach to investigate NUDT1 function if highly specific

and potent inhibitors are available. Please kindly note that several reported NUDT1 catalytic inhibitors, including TH588, exhibited limitations in terms of their selectivity and pharmacokinetic profiles. In this regard, PROTACs have a more accessible means for assessing proteome-wide selectivity through quantitative proteomics. Moreover, PROTACs possess the distinct advantage of inducing acute protein degradation, effectively mimicking gene knockdown or knockout experiments. Indeed, our PROTAC LC-1-40 displayed an excellent proteome-wide selectivity (revised Fig. 5g-j). Meanwhile, it also exhibited potent NUDT1 degradation activity, with a DC50 value of 0.97 nM and Dmax of 96% (revised Fig. 5c, d). We believe that LC-1-40 can serve as a superior probe for studying NUDT1 in comparison to traditional catalytic inhibitors.

Reviewer #2 (Remarks to the Author):

The manuscript by Ye, M., et. al. described NUDT1 dependency as a Myc-driven metabolic vulnerability by non-bias sg RNA screening. They provide convincing data showing that Myc activates NOX4 transcription and increases ROS. Since NUDT1 is a hydrolase of 8-oxo-dGTP, this study postulates that myc-induced ROS generates more 8-oxo-dGTP, thereby requiring NUDT1 for the sanitization of dNTPs and tumor growth. They also found that the Myc-PLK1 signal increases NUDT1 phosphorylation, which enhances the catalytic efficiency. Therefore, this signal adds a layer of NUDT1-mediated sanitization of dGTP in Myc-transformed cancer cells. Most importantly, they developed a potent PROTAC drug to degrade NUDT1, thus leading to myc-cancer cell death. Overall, this is a well-executed study for anti-cancer translational research. Several concerns regarding the claims stated in the manuscript are as followed.

We appreciate the Reviewer's expert comment.

Major comments:

1. The de novo synthesis of dNTPs requires RNR-mediated reduction of rNDP via NADPH, which relies on the redox cycle reaction of glutaredoxin/glutathione/glutathione reductase or thioredoxin reductase/thioredoxin/ FADH2. Data from Figure 2 provide convincing evidence that Myc directly activates NOX4 transcription. Since NOX4 upregulation leads to ROS induction with NADPH consumption, the outcome would be dNTP deficiency due to the requirement of NADPH and glutathione for RNR reaction. Then the question is whether dNTPs pools are affected by Myc-induced NOX4 even though Myc has been shown to upregulate the enzyme expression in nucleotide synthesis for cell proliferation.

The Reviewer raised an important scientific question. Based on the Reviewer's suggestion, we quantified dNTP abundance in SHEP MYCN-ER cells with or without 4-OHT treatment. As expected, 4-OHT induction of MYCN activation markedly increased the dNTP abundance in SHEP cells (revised Supplementary Fig. 2a), validating a critical role of MYC(N) in activating nucleotide biosynthesis. Most likely, cells still maintain enough NDAPH pools for nucleotide biosynthesis in spite of increased NADPH consumption due to NOX4 upregulation.

After all, the conversion of rNTP to dNTP still requires NADPH. Since the manuscript is focused

on the regulation of removal of 8-oxo-dGTP in myc-associated cancer, it would be important to determine whether dNTP pools are increased or reduced by Myc. If reduced, this would mean that a greater fraction of the dGTP pool contains 8-oxo-dGTP, thus making NUDT1-mediated deoxy-8-oxo-GTP hydrolysis necessary for avoiding 8-oxo-dGTP misincorporation and DNA damage-induced cell death. If this is the case, PLK1-mediated phosphorylation of NUDT1 might not play a great part in the orchestration of nucleotide sanitization for cell proliferation. Instead, it is Myc expression that markedly increases the 8-oxo-dGTP/dGTP ratio causing NUDT1 dependency.

As stated above, we quantified dNTP levels in SHEP MYCN-ER cells with or without 4-OHT treatment, and confirmed that 4-OHT induction of MYCN markedly increased the total dNTP (including dGTP) abundance (revised Supplementary Fig. 2a). Thus, the presence of supraphysiological concentrations of nucleotide pools imposes a more onerous cost on MYC(N)-overexpressing tumor cells to counteract their disastrous damages by elevated ROS due to NOX4 activation. We identify that these tumor cells strictly rely on NUDT1 to sanitize oxidized nucleotides for viability (revised Fig. 1h). We further show that PLK1-mediated S121 phosphorylation is crucial for NUDT1 catalytic activity (revised Fig. 3g, k). As such, in comparison to NUDT1 WT, the NUDT1 S121A mutant can only partially decreased the 8-oxo-dG levels due to depletion of endogenous NUDT1 upon 4-OHT induction of MYCN in SHEP MYCN-ER cells (revised Supplementary Fig. 4i, j). Based on the ternary structure of NUDT1 and 8-oxo-dGTP complex (Protein Data Bank code 5FSI), phosphorylation of S121 is likely to affect the configuration and flexibility of the substrate binding loop, which may accelerate the product-substrate exchange, and thereby enhancing the turnover of 8-oxo-GTP (revised Supplementary Fig. 4k). All these findings support that PLK1-directed S121 phosphorylation is an important mechanism in regulation of NUDT1 catalytic activity.

2. Figure S2 g showed that NUDT1 knockdown abolished 8-oxo-dG incorporation into DNA by avidin IF intensity in Myc-induced cells. If NOX4 is responsible for the increase in 8-oxo-dGTP, sgNOX4 should also reduce the basal level of 8-oxo-dGTP incorporation. However, the data revealed no difference.

We agree with the Reviewer that NOX4 depletion might reduce the basal level of 8-oxo-dGTP incorporation given the role of NOX4 in MYC activation of ROS production. To evaluate this possibility, we carefully repeated NOX4 knockdown and avidin immunofluorescence experiments in Kelly and SF188 cells. Consistent with our previous data, we did not observe a significant difference in 8-oxo-dGTP levels between control and sgNOX4 groups (revised Fig. 2h and Supplementary Fig. 2k). Most likely, when NUDT1 is present, the functionally active NUDT1 is capable of efficiently removing 8-oxo-dGTP in Kelly and SF188 cells such that depletion of NOX4 would not cause a significant alteration in basal 8-oxo-dGTP levels.

3. Figure 2 e should include a negative control for avidin readout by overexpression of OGG, which removes 8-oxo-G in DNA. Also peculiar is why NUDT1 knockdown still retained more 8-

oxo-G in DNA when considering that the removal of 8-oxo-G through BER and MMR in replication, which generate toxic DNA breaks and ultimately cell death.

We agree with the Reviewer that overexpression of OGG1 can be an ideal control for avidin readout. As expected, OGG1 overexpression effectively decreased the avidin immunofluorescence intensity (revised Fig. 2e).

We also agree with the Reviewer that, in addition to NUDT1, 8-oxo-dG can be removed by base excision repair (BER) and mismatch repair (MMR) pathways, which in principle would regulate 8-oxo-dG abundance and cell death. As OGG1 and MSH2 (MutS homolog 2) are critical BER and MMR executors, respectively, we first examined their expression in SHEP MYCN-ER model system. Interestingly, unlike MSH2, OGG1 was barely expressed in SHEP cells (revised Supplementary Fig. 3a), indicating BER was largely nonfunctional in this system. We then depleted MSH2 by a specific sgRNA. Of note, in comparison to NUDT1 depletion, inhibition of MSH2 alone did not induce dramatic SHEP cell death upon 4-OHT induction of MYCN hyperactivation, although it increased cell death further when NUDT1 is simultaneously depleted (revised Supplementary Fig. 3b), supporting that NUDT1 is a determining factor for MYC(N)-induced cell death.

Given the predominant role NUDT1 plays in MYC(N)-induced cell death, we speculate that the overwhelming nucleotide and DNA damages induced by NUDT1 depletion upon MYC(N) activation would override the cell's capacity to repair through MMR. To examine this, we performed a time course experiments in SHEP MYCN-ER cells by administration of the NUDT1 degrader LC-1-40. When MYCN is inactive (in the absence of 4-OHT), administration LC-1-40 caused an initial increase in avidin and γ H2AX staining in SHEP cells, but these signals gradually decreased as the treatment continued to 24 and 36 h. Most likely, MMR resolved these lesions at later time points. In sharp contrast, avidin and γ H2AX staining continued to increase upon 4-OHT induction of MYCN hyperactivation (data shown below, Fig. 1a, b). These data suggest that the excessive oxidized nucleotides override the capacity of MMR, leading to accumulation of lethal damages and ultimate cell death. In support of this notion, immunoblot assays confirmed that Caspase 3 was also continuously activated in LC-1-40 treated SHEP cells upon 4-OHT induction of MYCN activation (data shown below, Fig. 1c).

Fig. 1 (a) Immunofluorescence images of 8-oxo-dGTP incorporation in DNA and γ H2AX. Scale bar, 20 μ m. Quantifications of 8-oxo-dGTP fluorescence signals (n=10) and γ H2AX foci per cell are shown in **(b)**. **(c)** Immunoblots of γ H2AX and cleaved caspase 3 in SHEP MYCN-ER cells subjected to the indicated treatment, with Actin as a loading control.

4. Regarding Myc-induced NOX4 upregulation, cancer cell lines expressing high myc and low myc or tumor samples should be used to validate the correlation.

We analyzed NOX4 protein levels in multiple tumor cell lines and primary tumor samples, and found NOX4 expression was markedly elevated in MYC(N)-high tumor cells and tumor samples (revised Supplementary Fig. 2g). To evaluate whether this observation is representative of what occurs in human tumors, we analyzed neuroblastomas with and without MYCN amplification. NOX4 mRNAs were significantly elevated in the MYCN-amplified tumors when compared with non-amplified ones (revised Supplementary Fig. 2h). Moreover, expression between MYCN and NOX4 is significantly correlated in MYCN-amplified neuroblastoma samples (revised Supplementary Fig. 2i). All these data support that MYC(N)-induced NOX4 upregulation occurs in human tumors.

5. Figure 5 provides strong evidence that NUDT1 is phosphorylated by PLK1 at S121 residue. Since pS121 antibody works very well, again, it would be nice to find the correlation in cancer cell lines expressing high myc and low myc or tumor samples.

As expected, we observed elevated pS121 NUDT1 levels in MYC(N) high tumor cells and tumor samples in comparison to those with low MYC(N) expression (revised Supplementary Fig. 4h).

6. The contribution of enhancing NUDT1 catalytic function by PLK1-mediated phosphorylation in cells should be verified by examining the effect of re-expression of S121A mutant and WT NUDT1 on the level of 8-oxo-G-DNA or 8-oxo-dGTP in sgNUDT1-Myc-ER cells.

Based on the Reviewer's suggestion, we examined the effect of re-expression of the S121A mutant and WT NUDT1 on the level of 8-oxo-dGTP in sgNUDT1-MYCN-ER cells. In comparison to WT NUDT1, the S121A mutant can only partially decreased the 8-oxo-dG levels due to depletion of endogenous NUDT1 upon 4-OHT induction of MYCN in SHEP MYCN-ER cells (revised Supplementary Fig. 4i, j). Based on the ternary structure of NUDT1 and 8-oxo-dGTP complex (Protein Data Bank code 5FSI), phosphorylation of S121 is likely to affect the configuration and flexibility of the substrate binding loop, which may accelerate the product-substrate exchange, and thereby enhancing the turnover of 8-oxo-GTP (revised Supplementary Fig. 4k). All these findings support that PLK1-directed S121 phosphorylation is an important mechanism in regulation of NUDT1 catalytic activity.

7. A potent PROTAC drug, LC-1-40, against NUDT1 at sub-nano-molar range was developed in the study. The analysis of LC-1-40 was robustly performed. To assure the specificity, a dormant mutant should be employed for validation.

We appreciate the Reviewer's insightful suggestion. To further validate the LC-1-40 specificity, we generated an E77K dormant mutant, which retains the NUDT1 enzymatic activity but is resistant to LC-1-40 degradation (incapable of NUDT1 binding) (revised Fig. 5h, i). As expected, the E77K mutant rescued LC-1-40 induced cell death as effectively as the WT NUDT1 (revised Fig. 5j). All these data demonstrated that LC-1-40 is a highly potent, on-target NUDT1 degrader.

Minor Comments.:

1. The enzyme kinetic analysis reveals K_m and V_{max} changes by a phospho-mimetic mutant of NUDT1. Overall, the changes are moderate. Since the NUDT1 structure is available, it might be helpful to provide a molecular explanation for the phosphorylation effect on the catalytic efficiency based on structural information for explanation.

We appreciate the Reviewer's suggestion. Based on the ternary structure of NUDT1 and 8-oxo-dGTP complex (Protein Data Bank code 5FSI), S121 is a surface-exposed residue located to one of the substrate binding loops (revised Supplementary Fig. 4k). Phosphorylation of S121 is likely to affect the configuration and flexibility of the substrate binding loop, which may accelerate the product-substrate exchange, and thereby enhancing the turnover of 8-oxo-GTP.

2. CRBN contains stereoisomer forms, which can exert a stereospecific effect. Whether LC-1-40 is stereo-pure should be mentioned in the text.

We have clarified this point in the revised manuscript (see line 276 in red).

In summary, we hope that these changes have adequately addressed all the concerns and that the manuscript is now acceptable for publication in *Nature Communications*. Thank you very much for your time, effort, and great help in this matter.

Sincerely,

Guoliang Qing, Ph. D., Professor
Department of Cancer Biology
Medical Research Institute
Wuhan University, China

Point-by-point response to reviewer comments:

We would like to express our gratitude to the reviewers for their thorough and insightful comments on our manuscript. Addressing each comment has strengthened our study and clarified our manuscript. Our point-by-point responses are below, and updates in manuscript are in red text for ease of review.

REVIEWER COMMENTS

Reviewer #1 (Remarks to the Author):

The authors have been able to address most of my comments in the revised version of the manuscript. In my opinion, the manuscript is much improved and ready for publication.

Elmar Wolf

We appreciate the Reviewer's expert comment.

Reviewer #2 (Remarks to the Author):

This revised version provides convincing evidence that Myc-driven NOX4-ROS requires PLK1-regulated NUDT1 to sanitize dGTP for cell proliferation and prevent the toxicity of DNA damage. Most importantly, this research group developed a potent and specific PROTAC drug to degrade NUDT1 as a therapeutic agent for eradicating MYC-high tumors.

One concern remains for this revision regarding the result shown in supplementary Fig. 3b that MSH depletion increased cell death in NUDT1-depleted cells. It has been well established that MMR-mediated DNA repair in the removal of misincorporated nucleotides in DNA replication often leads to cell death due to futile repair involving EXOI processing. This is why MMR defect tumor is associated with 6-TG and 5FU resistance in anti-cancer treatment. Since OGG expression is very low, the presence of MMR would be beneficial for inducing cell death in NUDT1-depleted cells by causing more DNA breaks that result in a toxic repair effect.

We appreciate the Reviewer's expert comment.

REVIEWER #2 FURTHER COMMENTS

This revised version provides convincing evidence that Myc-driven NOX4-ROS requires PLK1-regulated NUDT1 to sanitize dGTP for cell proliferation and prevent the toxicity of DNA damage. Most importantly, this research group developed a potent and specific PROTAC drug to degrade NUDT1 as a therapeutic agent for eradicating MYC-high tumors. These merits deserve publication in Nature Communications.

We appreciate the Reviewer's expert comment.

However, there is still a concern about MSH depletion data provided in supplementary Fig. 3. The authors might misunderstand my original comment #3. What I meant is that the substrate of NUDT1 is the free form of 8-oxoGTP, which misincorporation into DNA can be recognized by BER or MMR system for removal. The removal of 8-oxoG in DNA without properly filling the gap thereby generates DNA breaks as revealed by γ -H2AX IF staining. The co-staining of avidin and γ H2ax in NUDT1 might indicate that the misincorporation and DNA breaks from the removal process co-exist, causing deterioration of DNA damage.

It has been well established that MMR-mediated DNA repair in the removal of misincorporated nucleotides in DNA replication often leads to cell death due to futile repair involving EXO1 processing. This is why MMR defect tumor is associated with 6-TG and 5FU resistance in anti-cancer treatment. Since OGG expression is very low, the presence of MMR would be beneficial for inducing cell death in NUDT1-depleted cells by causing more DNA breaks that result in a toxic repair effect. However, they found that MSH depletion indeed increased cell death in NUDT1-depleted cells (supplementary Fig 3b). The authors explained that MMR might still play a role in repairing 8-oxoG mis-incorporation. The result and explanation though are interesting but of concern. My suggestion is that they should add data from Fig.1 shown in the rebuttal letter and delete supplementary Fig. 3 because the explanation is not satisfactory.

Based on the Reviewer's insightful suggestion, we have removed supplementary Fig. 3 shown in the previous version and added Fig.1 in the rebuttal letter to the latest version (see Supplementary Fig. 8). We have also discussed this point in the revised manuscript (see lines 301-309 in red).

Point-by-point response to reviewer comments:

We would like to express our gratitude to the reviewers for their thorough and insightful comments on our manuscript. Addressing each comment has strengthened our study and clarified our manuscript. Our point-by-point responses are below, and updates in manuscript are in red text for ease of review.

Reviewer #2 (Remarks to the Author):

This revised version resolves the concern by providing data showing the sustained gH2AX staining in MYCN hyperactivation cells in response to LC-1-40 treatment, suggesting the dependency on NUDT1 in avoiding the accumulation of lethal DNA damage.

We appreciate the Reviewer's expert comment.